# Research

nanotechnology

nanocomposites, pod-shaped, TiO$_2$, morphology control, photocatalysis

**Author for correspondence:**
Zhiqiang Cheng
e-mail: czq5974@163.com

This article has been edited by the Royal Society of Chemistry, including the commissioning, peer review process and editorial aspects up to the point of acceptance.

$^†$These authors contributed equally to this work.

# Preparation of pod-shaped TiO$_2$ and Ag@TiO$_2$ nano burst tubes and their photocatalytic activity

Shang Wang$^{1,†}$, Zhaolian Han$^{1,†}$, Tingting Di$^2$, Rui Li$^1$, Siyuan Liu$^1$ and Zhiqiang Cheng$^1$

$^1$College of Resources and Environment, Jilin Agricultural University, Changchun 130118, People's Republic of China
$^2$Northeast Electric Power Design Institute Co., Ltd of China Power Engineering Consulting Group, Changchun 130021, People's Republic of China

ZH, 0000-0001-7238-9236; ZC, 0000-0003-1156-0540

The pod-shaped TiO$_2$ nano burst tubes (TiO$_2$ NBTs) were prepared by the combination of electrospinning and impregnation calcination with oxalic acid (H$_2$C$_2$O$_4$), polystyrene (PS) and tetrabutyl titanate. The silver nanoparticles (AgNPs) were loaded onto the surface of TiO$_2$ NBTs by ultraviolet light reduction method to prepare pod-shaped Ag@TiO$_2$ NBTs. In this work, we analysed the effect of the amount of oxalic acid on the cracking degree of TiO$_2$ NBTs; the effect of the concentration of AgNO$_3$ solution on the particle size and loading of AgNPs on the surface of TiO$_2$ NBTs. Scanning electron microscopy and transmission electron microscopy investigated the surface morphology of samples. X-ray diffraction and X-ray photoelectron spectroscopy characterized the structure and composition of samples. Rhodamine B (RhB) solution was used to evaluate the photocatalytic activity of pod-shaped TiO$_2$ NBTs and Ag@TiO$_2$ NBTs. The results showed that TiO$_2$ NBTs degraded 91.0% of RhB under ultraviolet light, Ag@TiO$_2$ NBTs degraded 95.5% under visible light for 75 and 60 min, respectively. The degradation process of both samples was consistent with the Langmuir–Hinshelwood first-order kinetic equation. Therefore, the catalytic performance of the sample is: Ag@TiO$_2$ NBTs > TiO$_2$ NBTs > TiO$_2$ nanotubes.

# 1. Introduction

Titanium dioxide (TiO$_2$) is a multifunctional semiconductor metal oxide, which has attracted an extensive range of research attention because of its unique optical, electronic and antibacterial properties. In practical applications, TiO$_2$ is not toxic, has

super-hydrophilicity and can completely contact with food [1] and does not affect human health. TiO$_2$ shows good development prospects in many fields. At present, TiO$_2$ has been widely used as a multifunctional material in the fields of solar cells [2], sensors [3], ceramics [4,5], especially photocatalysts [6–8] in many environmental pollution control programmes. Photocatalytic technology based on nano-TiO$_2$ materials provides a cheap, non-toxic, energy-efficient and highly efficient method for degrading organic pollutants in air and water [9–12]. As an indirect bandgap semiconductor material, TiO$_2$ has typical semiconductor energy band characteristics. The energy band consists of a valence band (VB) filled with electron orbits, an empty orbital conduction band (CB) without electrons and a band gap ($E_g$) between the VB and the CB. When TiO$_2$ is not excited, electrons in the VB do not automatically transition to the CB. Only when the energy excited by the photons is greater than $E_g$, the electrons in the VB absorb the energy of the photon transition into the CB, and holes are generated in the VB, 'electron–hole pairs' (e$^-$–h$^+$) are formed [13–16]. As the research groups continue to explore, the morphological structure of TiO$_2$ is also constantly changing. Nanorods, nanotubes, nanoflowers, nanoparticles and nanofibres have been studied and prepared. Although notable advances have been made, the high recombination rate of the photogenerated electron/hole pairs and the low utilization rate of ultraviolet hinder its further application in industry.

In order to improve the ability of TiO$_2$ to degrade organic pollutants, researchers have made many efforts. Coupling TiO$_2$ with other semiconductor or precious metal matrix composites or heterostructures provides a beneficial solution for defects in photocatalytic applications [17–20]; e.g. doping noble metals Au, Ag and Pt to increase the photocatalytic efficiency of TiO$_2$. Because the decorative precious metal forms a Schottky barrier with TiO$_2$, the separation of photogenerated carriers is significantly enhanced, and the interface electron transfer process is promoted. The interface charge transfer process in composite systems is promoted. Moreover, some precious metal nanoparticles (MNPs) can interact strongly with light in the visible region because of their extraordinary localized surface plasmon resonance (LSPR) characteristics [21–25]. This is due to the collective oscillation of electrons near the surface of the MNPs. At present, there are many studies on improving the photocatalytic activity of TiO$_2$ by using the LSPR characteristics of noble MNPs under visible light irradiation [26–28]. Among the precious metals, Ag is most suitable for industrial applications because of its low cost, non-toxicity and ease of preparation. The AgNPs are photochemically reduced by UV irradiation and deposited on the TiO$_2$ photocatalyst, which is a very simple method for synthesizing Ag@TiO$_2$ [29,30].

Here, we describe a method for preparing the novel pod-shaped TiO$_2$ nano burst tubes (NBTs) by electrospinning and impregnation calcination [6,31–34]. AgNPs were successfully loaded on the surface of TiO$_2$ NBTs by UV reduction to obtain Ag@TiO$_2$ NBTs. Scanning electron microscopy (SEM) and transmission electron microscopy (TEM) were used to observe surface morphology and size of the samples. The results of X-ray diffraction (XRD) and X-ray photoelectron spectroscopy (XPS) showed that the sample was mainly anatase TiO$_2$, and the TiO$_2$ NBTs were successfully modified after loading AgNPs. The DRS measurement results fully demonstrated that the addition of AgNPs improves the light absorption efficiency of the samples. The special cracking structure of pod-shaped TiO$_2$ NBTs and Ag@TiO$_2$ NBTs can effectively capture light and have a large surface area and more active sites [35]. Therefore, the sample could increase the rate of electron transfer process and reduce the recombination of charge carriers, thereby improving the catalytic ability of the sample. The current work provides a new form of support for metal and metal oxide particles, which has a high potential value for future photocatalytic research.

# 2. Material and methods

## 2.1. Materials

Polystyrene (PS) was provided by Shanghai Youngling Electromechanical Technology Co., Ltd (the PS with Mw = 150 000 g mol$^{-1}$). Tetrabutyl titanate (TBOT 97%), *N,N*-dimethylformamide (DMF), AgNO$_3$ and Rhodamine B (RhB) were purchased from Aladdin Industrial Corporation. Absolute ethanol and oxalic acid (H$_2$C$_2$O$_4$) were bought from Beijing chemical works. All chemicals were of analytic grade and used without further purification.

## 2.2. Preparation of pod-shaped TiO$_2$ NBTs

An amount of 1.15 g of PS was dissolved in 3.85 g of DMF, 0, 0.005, 0.015, 0.025 and 0.050 g of H$_2$C$_2$O$_4$ were added to the mixture (where the mass fraction of H$_2$C$_2$O$_4$ is 0%, 0.1%, 0.3%, 0.5% and 1.0%, respectively). Magnetic stirring was carried out for 6 h at 60°C. When the humidity of the

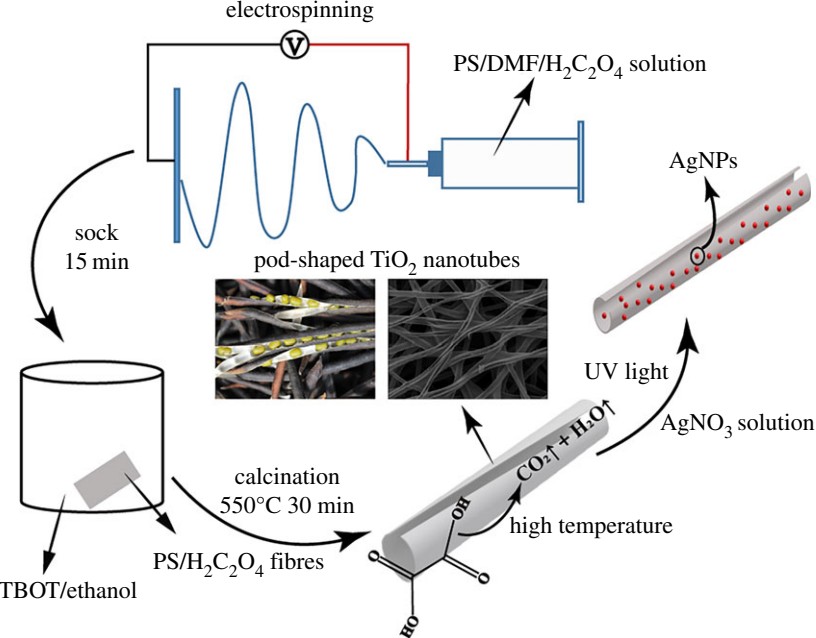

**Scheme 1.** The schematic of the synthesis process for pod-shaped Ag@TiO$_2$ NBTs.

environment was about 30%, the spinning solution was transferred to a syringe which was connected to a high voltage of 14 kV. The flow rate of the spinning solution was controlled at 1.0 ml h$^{-1}$, the distance between the needle and the rotating receiver was 16 cm and PS/H$_2$C$_2$O$_4$ fibre membrane was obtained. The fibre membrane was immersed in a TBOT/ethanol solution for 15 min, wherein the mass ratio of TBOT to ethanol solution was 1:10. When the fibres were completely infiltrated, they were taken out and placed in a 60°C oven for drying. Finally, the dried PS/H$_2$C$_2$O$_4$/TBOT fibre membrane was placed in a tube furnace and calcined to 550°C (electronic supplementary material, figure S1 shows why the calcination temperature is set to 550°C), wherein the heating rate was 3°C min$^{-1}$ and the holding time was 30 min. The pod-shaped TiO$_2$ NBTs with different morphology were obtained after annealing. The samples were marked for TiO$_2$(0%), TiO$_2$(0.1%), TiO$_2$(0.3%), TiO$_2$(0.5%) and TiO$_2$(1.0%).

## 2.3. Preparation of pod-shaped Ag@TiO$_2$ NBTs

The TiO$_2$(0.3%) sample was immersed in an AgNO$_3$ solution and magnetically stirred for 30 min in the dark with concentrations of AgNO$_3$ of 0.01, 0.05 and 0.1 M, respectively. The speed was set to just suspend the sample in the AgNO$_3$ solution to avoid damaging the sample. Finally, the mixture was placed under ultraviolet irradiation for 5 min and washed thoroughly with distilled water. Then, the composite was dried in an oven at 60°C to obtain pod-shaped Ag@TiO$_2$ NBTs, which were named (0.01 M)Ag@TiO$_2$(0.3%), (0.05 M)Ag@TiO$_2$(0.3%) and (0.10 M)Ag@TiO$_2$(0.3%), respectively. The synthesis process of pod-shaped Ag@TiO$_2$ NBTs is shown in scheme 1.

## 2.4. Catalyst characterization

The surface morphology of pod-shaped TiO$_2$ NBTs and Ag@TiO$_2$ NBTs was observed by SEM (SSX-550, Shimadzu) equipped with energy-dispersive X-ray analysis and TEM (Tecnai F20, FEI). The crystal structure properties of pod-shaped TiO$_2$ NBTs and Ag@TiO$_2$ NBTs were studied by X-ray diffractometry (XRD, XRD-7000, Shimadzu) with Cu Kα (λ = 0.15418 nm) as radiation source and the scanning range from 20° to 80°. The thermogravimetric differential thermal curves (TG-DTA, HCT-3, Beijing Henven Scientific Instrument Factory) of the H$_2$C$_2$O$_4$ and precursor nanofibres were analysed. The specific surface area of the sample was measured by specific surface and pore size analysis instrument (3H-2000PS1, BeiShiDe instrument). XPS was performed on a VG ESCALAB LKII instrument with Mg-Kα-ADES (hv = 1253.6 eV) source at a residual gas pressure of below 1028 Pa. The optical properties of samples were evaluated using a diffuse reflectance ultraviolet–visible

spectrophotometer (DRS, TU-1950, Beijing Purkinje General) with $BaSO_4$ as reference. The absorbance of RhB at 554 nm was measured by ultraviolet–visible spectrophotometer.

## 2.5. Photocatalytic activity measurement

By using the degradation rate of RhB solution, the photocatalytic activity of pod-shaped $TiO_2$ NBTs was investigated. To ensure the mixture reached the adsorption–desorption equilibrium, the mixture of RhB and samples was magnetically stirred in the dark for 2 h. The mixture was exposed to irradiation of high-pressure mercury lamp (360 W). The 5 ml solution was extracted from the mixture every 15 min and centrifuged to determine the concentration of RhB in the solution by UV–visible spectrum.

The photocatalytic experimental procedure of three samples of pod-shaped $Ag@TiO_2$ NBTs was the same as the above process, and the $TiO_2(0.3\%)$ as a blank comparison. The difference was that the visible light was irradiated with 360 W, and the sampling interval was 10 min.

# 3. Results and discussion

Figure 1 shows SEM images of five samples, TEM and high-resolution TEM (HRTEM) images of the sample $TiO_2(0.3\%)$ and the XRD patterns of five samples. According to figure 1$a$, it can be known that when the mass fraction of $H_2C_2O_4$ is 0%, PS and DMF were used to configure the spinning solution, and the diameter of $TiO_2(0\%)$ is about 0.8–1.5 µm. As shown in figure 1$b$–$d$, the diameter of the sample was gradually reduced as $H_2C_2O_4$ was added; the diameter range was maintained at 0.6–1.2 µm. It can be clearly seen from figure 1$a$–$e$ that the morphology of the $TiO_2$ NBTs has changed due to the difference in the amount of $H_2C_2O_4$ added. The greater the amount of $H_2C_2O_4$ added, the greater the degree of cracking of the $TiO_2$ NBTs. It can be clearly seen from the illustration in figure 1$c$ that the crack width of the $TiO_2(0.3\%)$ is 70.0% at 200–400 nm. The $TiO_2(1.0\%)$ has been transformed from a tubular shape to a sheet shape with a width of 0.7–2.4 µm from figure 1$e$, so $TiO_2$ is not only a planar structure in space but also a pleated form. In summary, the cracking process of $TiO_2$ NBTs is very similar to the ripening process of pod. Therefore, it was described as pod-shaped $TiO_2$ NBTs. As shown in figure 1$f$, the TEM image of sample $TiO_2(0.3\%)$, the presence of many pore structures can clearly be observed. Figure 1$g$ is an HRTEM image of $TiO_2(0.3\%)$, and the lattice spacing $d$ of the sample was measured to be 0.351 nm.

Figure 1$h$ depicts the XRD patterns of $TiO_2(0\%)$, $TiO_2(0.1\%)$, $TiO_2(0.3\%)$, $TiO_2(0.5\%)$ and $TiO_2(1.0\%)$. The main diffraction peaks of $TiO_2$ at 25.28°, 37.80°, 48.05°, 53.89°, 55.06° and 62.69° can be indexed to (101), (004), (200), (105), (211) and (204) direction, which characterizes the anatase structure of $TiO_2$ (JCPDS 21-1272). The 25.28° diffraction peak is sharpest and points to the anatase $TiO_2$ (101) crystal plane. The lattice spacing of the anatase $TiO_2$ (101) crystal plane is 0.351 nm, which is the same as the HRTEM image measurement. The main diffraction peaks of $TiO_2$ at 27.45° and 36.09° are indexed to the (110) and (101) directions, which characterize the rutile-type $TiO_2$ (JCPDS 21-1276). The diffraction peaks in the (110) and (101) directions are considered as by-product peaks. The peak of the XRD pattern corresponding to the $TiO_2$ rutile structure is very small, indicating that the purity of the sample is acceptable.

The result of TG-DTA analysis of pure oxalic acid and PS/TBOT composite fibre samples with $H_2C_2O_4$ content of 0.3% is shown in figure 2. As can be seen from figure 2$a$, the complete decomposition temperature of pure $H_2C_2O_4$ is 250°C, and $H_2C_2O_4$ has been completely decomposed when PS has not been decomposed. Figure 2$b$ shows that the thermal decomposition process of the sample was divided into three stages. The first stage occurred between 70 and 350°C. The sample lost water and $H_2C_2O_4$ decomposed to release $CO_2$ gas. The second stage is 350–450°C, which represents the decomposition of PS organic components and some of the organics produced by TBOT hydrolysis. The third stage is 450–700°C, mainly including PS main chain degradation and amorphous $TiO_2$ to anatase phase 2 processes.

As shown in figure 3$a$, the specific surface areas of the samples were 28.16, 65.35, 67.55, 43.42 and 51.89 $m^2\,g^{-1}$. Sample $TiO_2(0.3\%)$ has large specific surface area, which may be due to 0.3% $H_2C_2O_4$ in the sample. In the calcining process, $H_2C_2O_4$ decomposes to release $CO_2$ gas and water vapour at high temperature, which thins the tube wall and leaves holes, causes the tube wall to break under the impact of a large amount of gas. All samples exhibited the same hysteresis loop in the range of 0.3–1.0 $P/P_0$, indicating a similar pore structure between samples. Figure 3$b$–$f$ shows the BJH pore size distribution of five samples, and the porous structure of the sample was investigated. The total pore volumes of $TiO_2(0\%)$, $TiO_2(0.1\%)$, $TiO_2(0.3\%)$, $TiO_2(0.5\%)$ and $TiO_2(1.0\%)$ were 0.14, 0.36, 0.52, 0.25 and 0.23 $cm^3\,g^{-1}$, respectively. The average pore diameters were 14.05, 16.12, 25.39, 13.74 and 14.57 nm, respectively. The $TiO_2(0.3\%)$ NBTs (figure 3$d$) have the largest total pore volume and

**Figure 1.** SEM images of five samples: (a–e) were sampled at TiO$_2$(0%), TiO$_2$(0.1%), TiO$_2$(0.3%), TiO$_2$(0.5%) and TiO$_2$(1.0%), respectively (the illustration in (c) shows the sample crack width statistics), (f) TEM image and (g) HRTEM image of sample TiO$_2$(0.3%) and (h) the XRD patterns of the five samples.

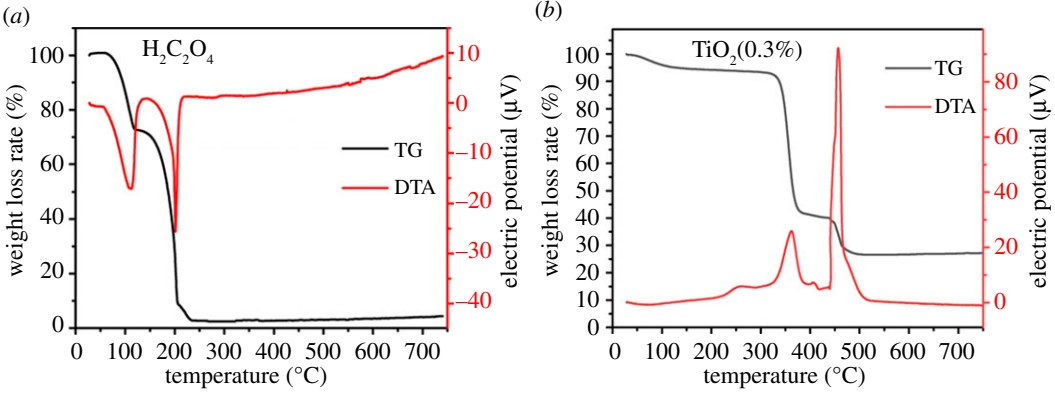

**Figure 2.** TG-DTA analysis of (*a*) H$_2$C$_2$O$_4$ and (*b*) TiO$_2$(0.3%) precursor fibre soaked in TBOT.

average pore size, while having the largest specific surface area. It also showed excellent performance in photocatalytic performance testing.

The morphology of samples TiO$_2$(0.3%), (0.01 M)Ag@TiO$_2$(0.3%), (0.05 M)Ag@TiO$_2$(0.3%) and (0.10 M)Ag@TiO$_2$(0.3%) was also characterized by SEM and TEM (the EDS spectrum of the loaded silver nanoparticle sample is shown in electronic supplementary material, figure S2). Figure 4*a* shows the morphology of the TiO$_2$ NBTs when the amount of H$_2$C$_2$O$_4$ is 0.3%. Figure 4*b*–*d* shows images of AgNPs deposited on the surface of pod-shaped TiO$_2$ NBTs when the concentration of AgNO$_3$ solution is 0.01, 0.05 and 0.10 M, respectively. It can be seen from the images that by controlling the concentration of AgNO$_3$ solution, effect the deposition amount of AgNPs on the surface of the pod-shaped TiO$_2$ NBTs. From the TEM image of the sample (0.05 M)Ag@TiO$_2$(0.3%) (figure 4*e*), the presence of AgNPs on the surface of sample could not be clearly observed. Therefore, an HRTEM image of the sample is also required to demonstrate the growth of AgNPs on the surface of the pod-shaped TiO$_2$ NBTs. The lattice spacing of 0.237 and 0.351 nm was found in figure 4*f*, thus demonstrating the existence of AgNPs.

Figure 5 shows an XRD pattern of pod-shaped TiO$_2$(0.3%) and (0.05 M)Ag@TiO$_2$(0.3%) samples. The crystal structure of a specific sample TiO$_2$(0.3%) has been seen in figure 1*h*. Therefore, the diffraction peak corresponding to Ag is mainly analysed in figure 5. Because the deposition amount of AgNPs was too low to be detected, the Ag peak was not clearly observed at the position of 38.1°, 44.3° and 64.5°. Therefore, XPS analysis of the sample is required to further demonstrate the presence of silver nanoparticles in the sample.

XPS analysis was used to determine the chemical composition and bonding configuration of the prepared samples. The entire spectrum of TiO$_2$(0.3%) and (0.05 M)Ag@TiO$_2$(0.3%) (0–1000 eV) is shown in figure 6*a*. According to figure 6*b,c*, it can be concluded that the Ti 2p3/2 and Ti 2p1/2 binding energies of the sample TiO$_2$(0.3%) are 458.38 and 464.16 eV, respectively. When AgNPs are deposited onto the TiO$_2$(0.3%) by photoreduction, the binding energies of Ti 2p3/2, Ti 2p1/2 and peaks were converted to 458.57 and 464.28 eV, respectively. At the same time, the peak value of O 1s increased from 529.55 to 529.75 eV. These data are significantly higher than the corresponding values for sample TiO$_2$(0.3%). This transformation means that there is an electron interaction between the sample TiO$_2$(0.3%) and Ag. Therefore, the binding energy of Ti 2p and O 1s increases with the increase in the lower Fermi level of Ag, and the electrons on the surface of TiO$_2$ will be transferred to AgNPs, resulting in a decrease in the electron cloud density of Ti ions. Figure 6*d* provides a high-resolution XPS spectrum of Ag in sample (0.05 M)Ag@TiO$_2$(0.3%) with a peak of 367.48 eV at Ag 3d5/2 and a standard value of 368.2 eV for bulk metal Ag. The binding energy of AgNPs was significantly reduced. The electrons of AgNPs are transferred to the surface of TiO$_2$, thereby reducing the electron cloud density of silver ion. Therefore, XPS analysis fully demonstrates the existence of Ag nanoparticles.

As we all know, the photocatalytic efficiency of photocatalytic materials is closely related to the light absorbance and wavelength range. The light absorption characteristics of the sample under different wavelengths of ultraviolet light can be detected by ultraviolet–visible diffuse reflectance spectroscopy (DRS). It can be seen from figure 7*a* that the adsorption spectra of the TiO$_2$(0.3%) and (0.05 M)Ag@TiO$_2$(0.3%) are different. (0.05 M)Ag@TiO$_2$(0.3%) has a broad absorption spectrum in the range of 250–600 nm, and its ability to absorb light is stronger than that of sample TiO$_2$(0.3%). By loading AgNPs, the absorbance in the visible region increases, while increasing the photoactivity of

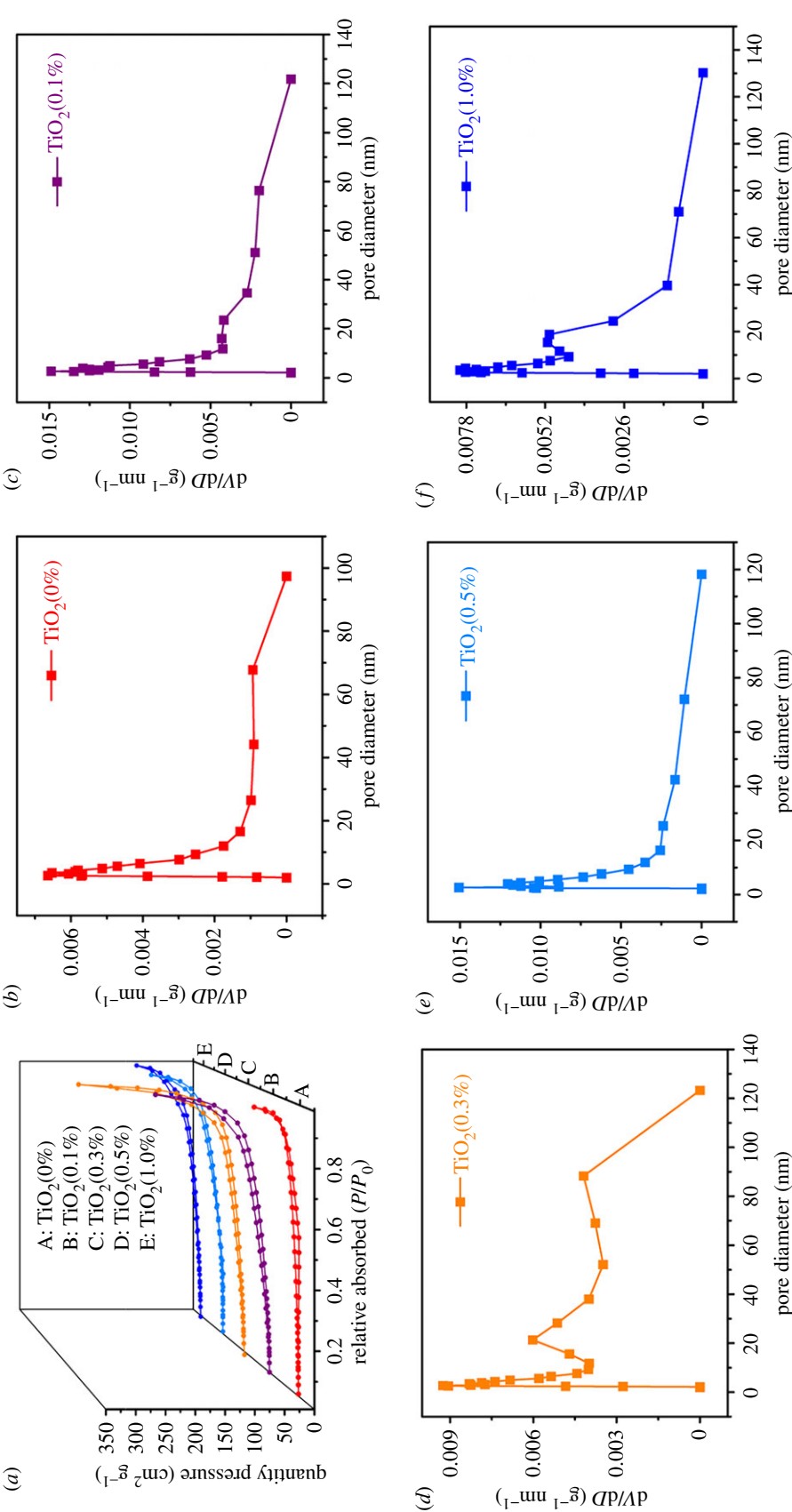

**Figure 3.** (a) Nitrogen sorption–desorption isotherms of the samples and (b–f) BJH pore size distribution of five samples.

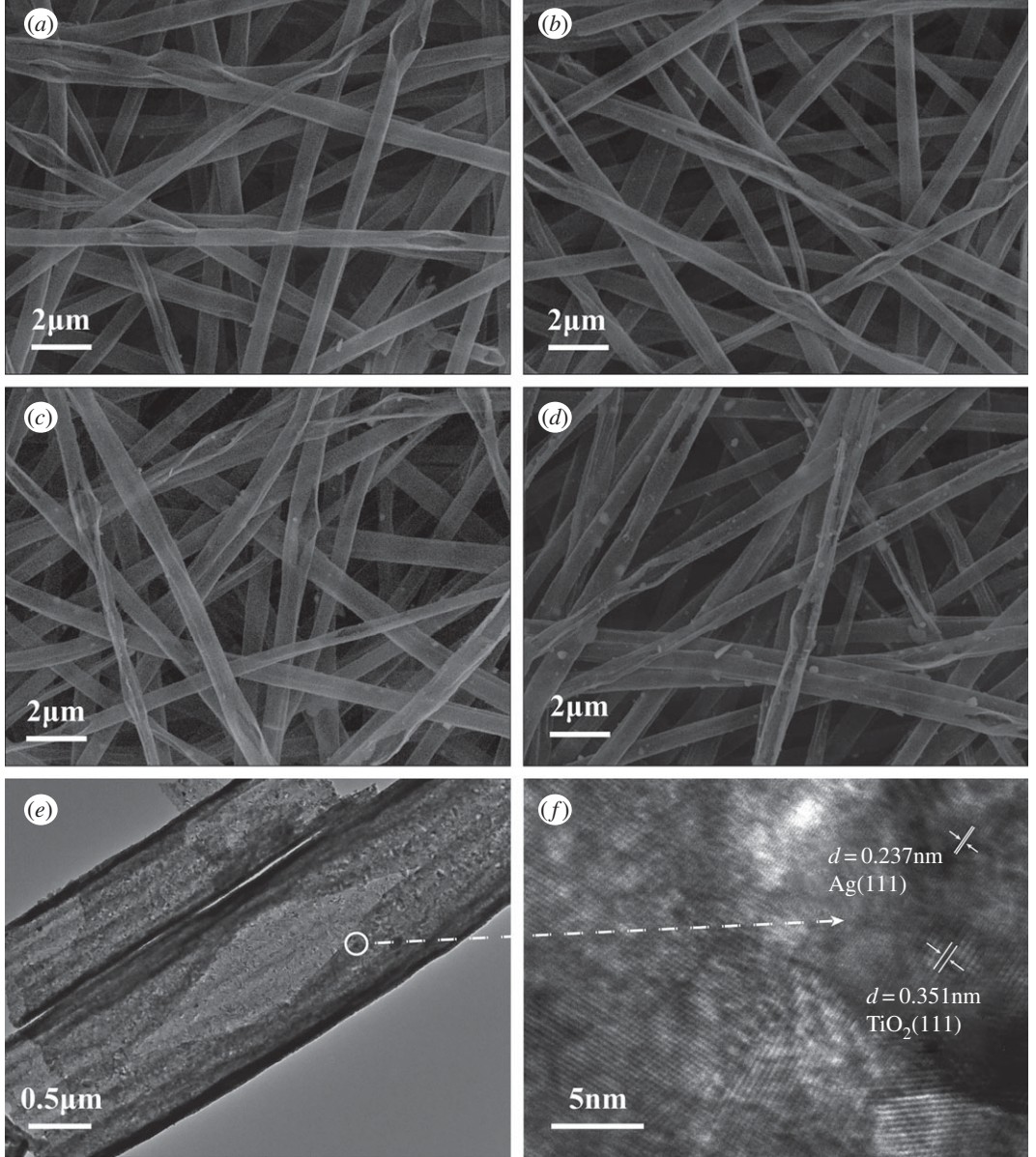

**Figure 4.** SEM images of four samples: (*a–d*) were sampled at TiO$_2$(0.3%), (0.01 M)Ag@TiO$_2$(0.3%), (0.05 M)Ag@TiO$_2$(0.3%) and (0.10 M)Ag@TiO$_2$(0.3%), respectively; (*e*) TEM image and (*f*) HRTEM image of sample (0.05 M)Ag@TiO$_2$(0.3%).

the sample under visible light. Because AgNPs are dispersed on the pod-shaped TiO$_2$ NBTs, under visible light irradiation, electrons in AgNPs are excited by light to form LSPR effect, which is more conducive to increasing visible light activity. The semiconductor forbidden band width was determined by the equation $E_g$ (eV) = 1240/$\lambda_g$ (absorption threshold, $\lambda_g$ (nm)) [36], the tangent axes of the TiO$_2$(0.3%) and (0.05 M)Ag@TiO$_2$(0.3%) were 424.66 and 454.21 nm, respectively, and the forbidden band widths ($E_g$) of the two samples were 2.92 and 2.73 eV, respectively. The absorption light range of TiO$_2$(0.3%) loaded with silver nanoparticles moves in the visible light direction. Therefore, the sensitivity of sample (0.05 M)Ag@TiO$_2$(0.3%) to light is enhanced. Figure 7*b* shows the Tauc plot [37] of TiO$_2$(0.3%) and (0.05 M)Ag@TiO$_2$(0.3%). According to the tangent position in figure 7*b*, the band gap energy of TiO$_2$(0.3%) also was estimated to be 2.92 eV, and the (0.05 M)Ag@TiO$_2$(0.3%) band gap energy of deposited AgNPs was 2.73 eV. Therefore, figure 7 shows the absorption light range of TiO$_2$(0.3%) loaded with AgNPs moves towards the visible light direction, and the forbidden band width is reduced, which improves the light utilization efficiency of the sample.

As shown in figure 8*a*, the mixed solution of RhB and five samples degrades under the irradiation of high-pressure mercury lamp. Before the catalytic degradation experiment, the solution containing 10 mg of different nanofibres was magnetically stirred for 2 h in a dark environment to reach an adsorption–

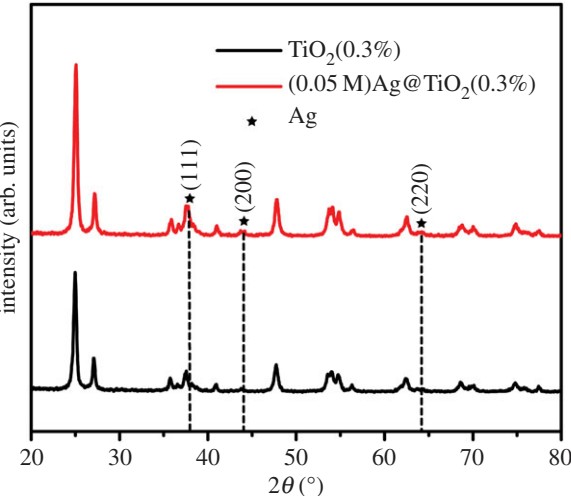

**Figure 5.** The XRD patterns of the samples TiO$_2$(0.3%) and (0.05 M)Ag@TiO$_2$(0.3%).

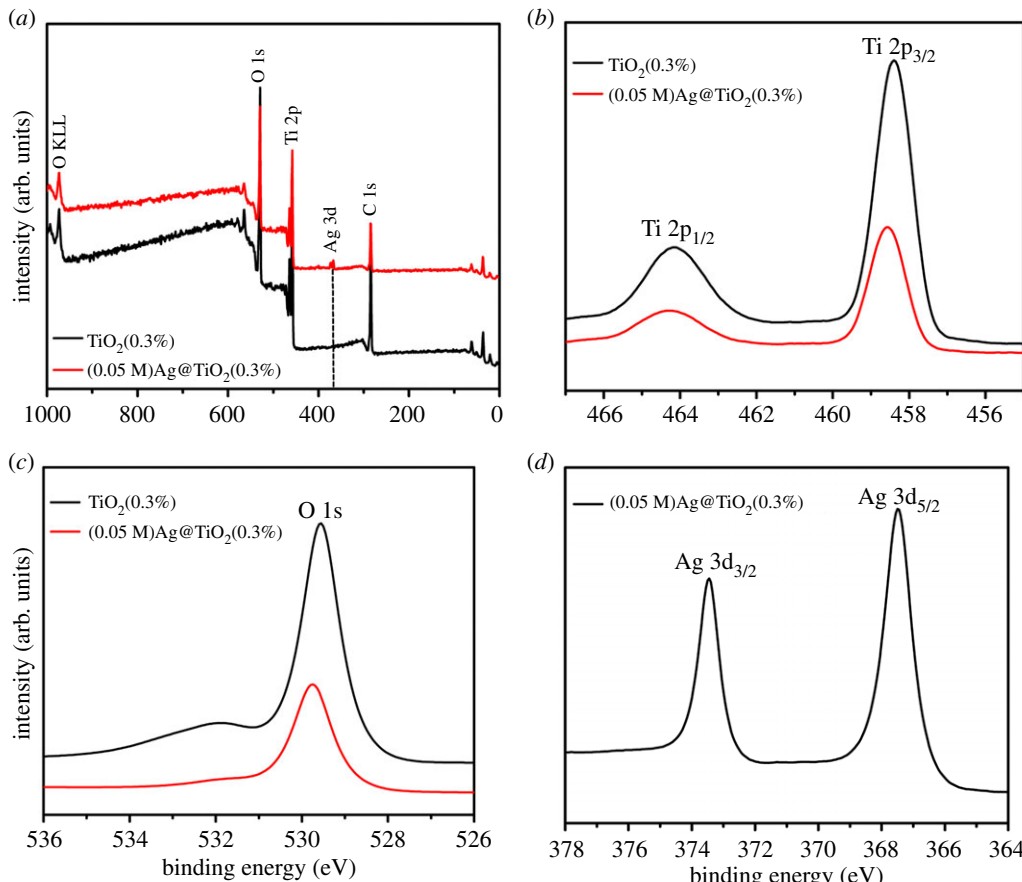

**Figure 6.** XPS spectra of sample TiO$_2$(0.3%) and (0.05 M)Ag@TiO$_2$(0.3%). (*a*) TiO$_2$(0.3%) and (0.05 M)Ag@TiO$_2$(0.3%) for complete measurement spectra, (*b*) Ti 2p peak, (*c*) O 1s peak and (*d*) Ag 3d peak.

desorption equilibrium. $C_0$ and $C_t$ are the initial concentration of RhB solution (10 mg l$^{-1}$, 10 ml) and the concentration at time $t$. It can be seen from figure 8*b* that the degradation process of RhB conforms to the Langmuir–Hinshelwood first-order kinetic reaction for the low-concentration substrates, and the kinetics equation can be expressed as: $\ln C_0/C_t = kt$ (the degradation rate constants ($k$, min$^{-1}$) were 0.3677, 0.3149, 0.7277, 0.3320 and 0.1708 for the five samples). The TiO$_2$(0.3%) has the highest degradation rate constant, which directly indicates that TiO$_2$(0.3%) has the best catalytic performance.

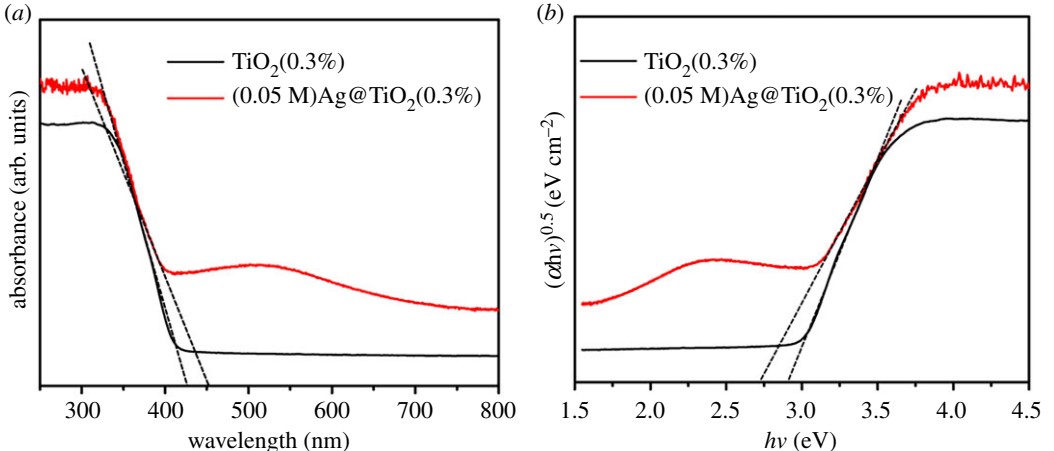

**Figure 7.** UV–Vis diffuse reflectance spectra (*a*) and Tauc plot (*b*) of samples TiO₂(0.3%) and (0.05 M)Ag@TiO₂(0.3%).

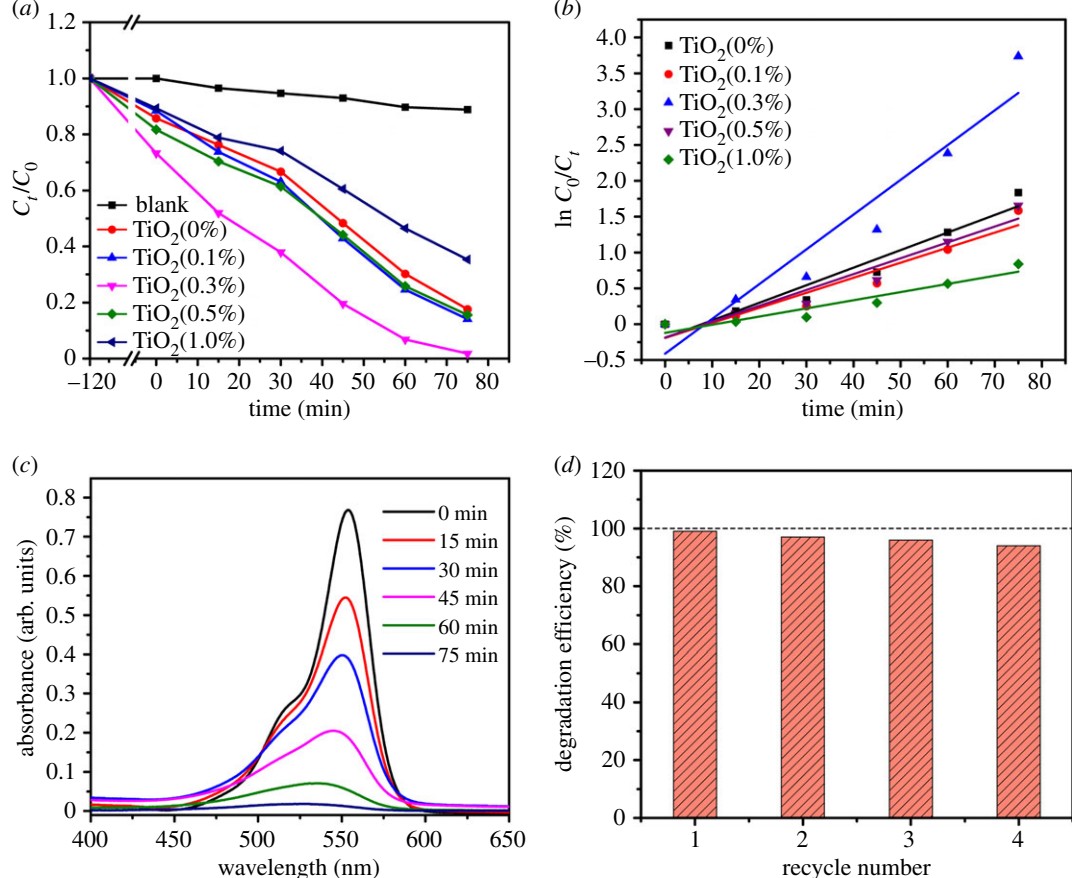

**Figure 8.** (*a*) Photocatalytic activities of samples photocatalyst on photodegradation of RhB under ultraviolet-light irradiation and (*b*) the corresponding kinetic studies; (*c*) UV–Vis absorption spectrum of degradation of RhB with sample TiO₂(0.3%); (*d*) the repeatability of four cycles of TiO₂(0.3%) test.

Absorption spectra of the solution were measured by UV–Vis spectroscopy, and RhB solution had a strong absorption peak at 554 nm (figure 8*c*). When the catalytic time was 60 min, the sample TiO₂(0.3%) had the best catalytic effect and successfully degraded 91.0% of RhB. When determining the surface area of the sample, the surface area of TiO₂(0.3%) is the largest of all samples, which also means that the TiO₂ surface has more pores and active sites. Thus, many nanocrystals and interfaces can further promote the diffusion of photogenerated electrons/holes throughout the structure, and the unique pod-shaped TiO₂ NBTs can provide an additional route for the light generated by the degradation process to generate

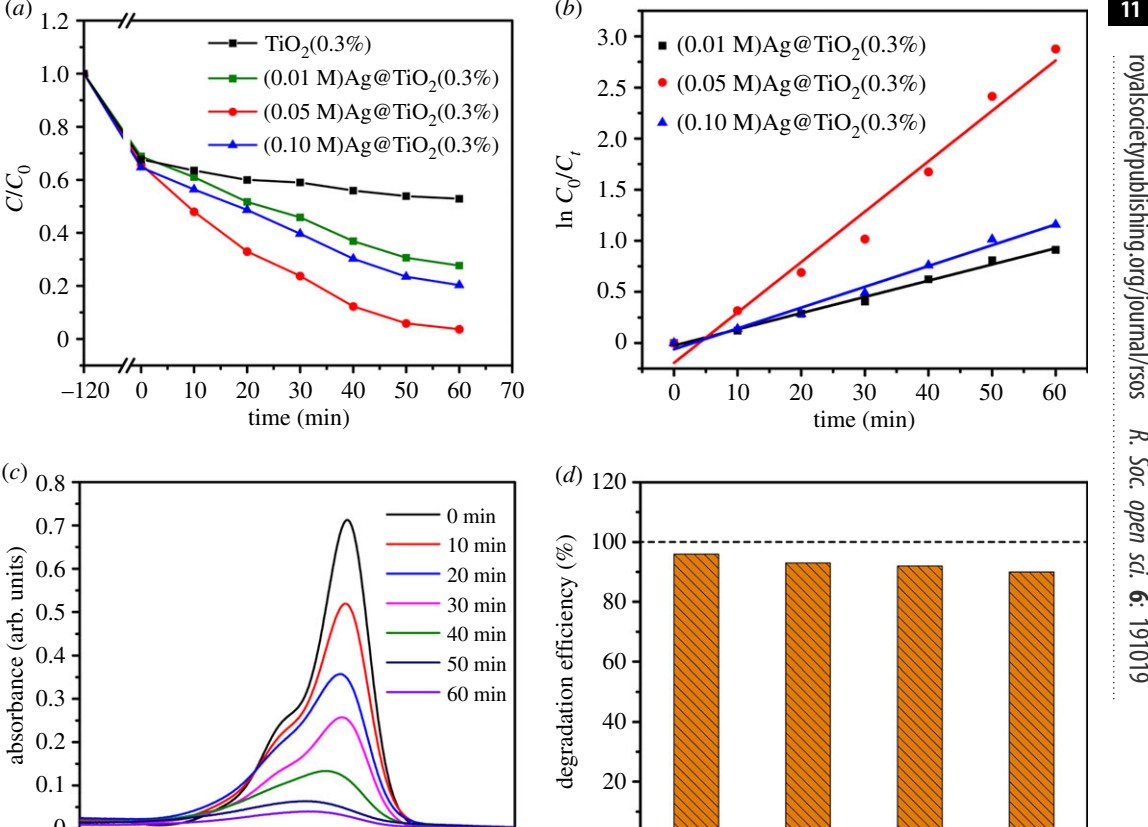

**Figure 9.** (a) Photocatalytic degradation rate of the sample under visible light and (b) kinetic studies; (c) UV–Vis absorption spectrum of the sample (0.05 M)Ag@TiO$_2$(0.3%) degrading RhB; (d) the four-cycle repeatability test of the sample (0.05 M) Ag@TiO$_2$(0.3%).

charge. In addition, figure 8d shows the degradation efficiency of the TiO$_2$(0.3%) recovered four times. After four reuses, the degradation efficiency of sample TiO$_2$(0.3%) was 94.0%. The photocatalytic activity of sample decreased with the increase in the number of times of recovery. The reason may be that the morphology of pod-shaped TiO$_2$ NBTs changes as the degradation experiment proceeds. In the case of multiple agitation, TiO$_2$ is broken down from pod-shaped into sheet shape.

In order to increase the photocatalytic activity of TiO$_2$(0.3%), pod-shaped TiO$_2$ NBTs were prepared by using three different concentrations of AgNO$_3$ solution. The sample degraded the RhB solution under irradiation with visible light. Figure 9a shows the concentration change curve of RhB. Figure 9b shows that (0.01 M)Ag@TiO$_2$(0.3%), (0.05 M)Ag@TiO$_2$(0.3%) and (0.10 M)Ag@TiO$_2$(0.3%) were also consistent with the first-order kinetic equation; therefore, the degradation rate constants ($k$, min$^{-1}$) of the three samples were 0.159, 0.0493 and 0.0204, respectively.

Figure 9c shows the UV–Vis absorption spectrum of the degradation of RhB by sample (0.05 M)Ag@TiO$_2$(0.3%) alone. The results showed that the photocatalytic degradation efficiency of sample (0.05 M)Ag@TiO$_2$(0.3%) was much higher than that of TiO$_2$(0.3%), (0.01 M)Ag@TiO$_2$(0.3%) and (0.10 M)Ag@TiO$_2$(0.3%) after 60 min irradiation with visible light. Among them, the degradation efficiency of (0.05 M)Ag@TiO$_2$(0.3%) was 95.5%, while the degradation efficiency of other samples was 23.5, 59.4 and 69.2%. (0.05 M)Ag@TiO$_2$(0.3%) showed the best degradation effect. Meanwhile, the degradation efficiency of the sample recovered four times is shown in figure 9d, and as the number of recoveries increases, the degradation efficiency of the (0.05 M)Ag@TiO$_2$(0.3%) sample also decreases. After four reuses, the degradation efficiency of the sample was 90%. The reason is that after the sample is recovered, AgNPs may be detached from the surface of the sample TiO$_2$(0.3%) and the morphology of pod-shaped TiO$_2$ NBTs is also changing, which affects the catalytic effect.

The SEM image of pod-shaped TiO$_2$(0.3%) and (0.05 M)Ag@TiO$_2$(0.3%) NBTs before and after the catalytic experiment is shown in electronic supplementary material, figure S3.

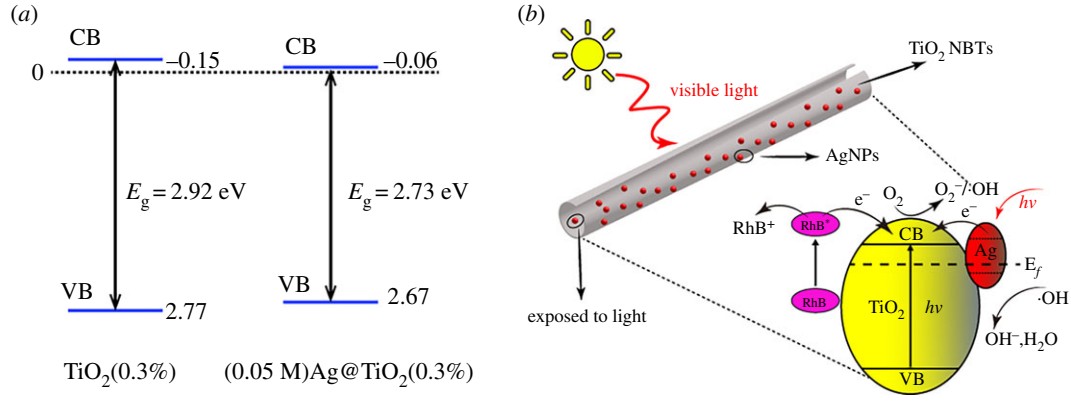

**Figure 10.** (*a*) Band edge (VB, CB) position of TiO$_2$(0.3%) and (0.05 M)Ag@TiO$_2$(0.3%) NBTs; (*b*) schematic diagram of charge separation and transfer of pod-shaped Ag@TiO$_2$ NBTs under visible light irradiation.

Figure 10*a* shows band edge (VB, CB) position of TiO$_2$(0.3%) and (0.05 M)Ag@TiO$_2$(0.3%) NBTs. Based on the bandgap energy obtained in figure 7, the empirical equation is used to calculate the position of the CB and the VB in the sample: $E_{CB} = X - E_e - E_g/2$ and $E_{VB} = E_{CB} + E_g$ [38], where $E_{CB}$ is the CB edge potential, $E_{VB}$ is the VB edge potential, and $X$ is the electronegativity of the semiconductor, which is the arithmetic mean of the electronegativity of constituent atoms and first ionization energy. $E_e$ is the energy of free electrons on the hydrogen scale (approx. 4.5 eV), and $E_g$ is the band gap energy of the semiconductor. The band edge positions (VB, CB) of TiO$_2$(0.3%) were 2.77 and −0.15 eV; the band edge positions of (0.05 M)Ag@TiO$_2$(0.3%) were 2.67 and −0.06 eV. According to the band edge position (VB, CB) of TiO$_2$(0.3%) and (0.05 M)Ag@TiO$_2$(0.3%), it can be known that when the AgNPs are deposited on the surface of TiO$_2$, the forbidden band width and band edge of the sample are effectively reduced. Therefore, (0.05 M)Ag@TiO$_2$(0.3%) has a good catalytic effect.

The transfer behaviour of charge during degradation is shown in figure 10*b*. The reaction equation is as follows:

$$RhB + visible\ light \rightarrow RhB^*, \tag{3.1}$$

$$RhB^* + TiO_2 \rightarrow RhB^+ + TiO_2(e^-), \tag{3.2}$$

$$Ag + visible\ light \rightarrow Ag(e^-) + Ag(h^+), \tag{3.3}$$

$$Ag(e^-) + TiO_2 \rightarrow TiO_2(e^-), \tag{3.4}$$

$$TiO_2(e^-) + O_2 \rightarrow \cdot O_2^-, \tag{3.5}$$

$$\cdot O_2^- + e^- + H^+ \rightarrow \cdot OH + OH^-, \tag{3.6}$$

$$Ag(h^+) + H_2O \rightarrow \cdot OH, \tag{3.7}$$

$$Ag(h^+) + OH^- \rightarrow \cdot OH \tag{3.8}$$

and

$$RhB^+ + \cdot OH \rightarrow CO_2 + H_2O. \tag{3.9}$$

RhB can be excited by visible light [39]. When Ag@TiO$_2$ nanocomposites are used as photocatalysts, the RhB molecule was adsorbed onto the surface of the nanocomposite. As shown in figure 10*b*, RhB* excited by visible light irradiation injects photoelectrons into the CB of the anatase phase, and the RhB molecule is converted to RhB$^+$ and undergoes further reaction to form the final product (equations (3.1), (3.2) and (3.9) are used as references). On the other hand, under visible light illumination, electrons in AgNPs are excited by light to form an LSPR effect, which results in visible light activity. At the same time, since Ag has a large work function, a Schottky barrier is formed at the Ag@TiO$_2$ interface. AgNPs absorb visible light and generate photogenerated electron/hole pairs, the generated electrons are transferred to the CB of the anatase phase (equations (3.4) and (3.5)). Obviously, the reaction increases the electron transport path, which also prevents the recombination of photogenerated electrons and holes. At the same time, the degradation efficiency of RhB is also improved. Through the above-described transfer behaviour of photogenerated electrons, holes accumulate on the surface of AgNPs. Photogenerated electrons accumulate in the CB of the anatase phase and participate in the formation of superoxide anion radicals ($\cdot O_2^-$), which combine with H$^+$ and e$^-$ to form hydroxyl radicals ($\cdot OH$) and hydroxide ions (OH$^-$) (equations (3.5) and (3.6)). The holes (h$^+$) accumulated on the surface of AgNPs react with H$_2$O and OH$^-$ in the RhB solution to form hydroxyl radicals ($\cdot OH$) (equations (3.7) and (3.8)). Finally, RhB$^+$ reacts with hydroxyl radicals ($\cdot OH$) to form environmentally friendly CO$_2$ and H$_2$O (equation (3.9)).

# 4. Conclusion

This paper proposes a new method for preparing pod-shaped $TiO_2$ NBTs by electrospinning technology and impregnation calcination. AgNPs were successfully loaded on the surface of $TiO_2$ NBTs by photoreduction to obtain Ag@$TiO_2$ NBTs. The special cracking structure of pod-shaped $TiO_2$ NBTs and Ag@$TiO_2$ NBTs can effectively capture light and have a large surface area and more active sites. At the same time, the process of catalyst degradation of RhB is consistent with the first-order kinetic reaction of the Langmuir–Hinshelwood low-concentration substrate. They exhibited an excellent capacity of catalytic degradation of RhB solution. The current work provides a new form of carrier for metal and metal oxide particles, which has the high potential value for future photocatalytic research.

Data accessibility. Data available from the Dryad Digital Repository: https://doi.org/10.5061/dryad.bd74r0t [40].

Authors' contributions. S.W. made substantial contributions to conception, design, analysis, interpreted data and drafted the manuscript; Z.H. participated in data analysis and the design of the study; T.D., R.L. and S.L. carried out the statistical analyses and collected field data; Z.C. conceived of the study, designed the study and critically revised the manuscript. All authors gave final approval for publication.

Competing interests. We have no competing interests.

Funding. This work was funded by Jilin Province Innovation Capacity Building Funds Project (2019IC0050-9); Jilin Province Key Technology R&D Project (20180101212NY) and Changchun Science and Technology Project (18DY023).

Acknowledgements. This paper was made better thanks to helpful comments from Shengzhe Zhao, Mingxu Chu, Chaoyue Zhao, Caijiao Ying and Yafeng Yuan.

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
