## [Reviewer comments · Royal Society Open Science]

Review History

RSOS-191019.R0 (Original submission)

Review form: Reviewer 1

Is the manuscript scientifically sound in its present form?

No

Are the interpretations and conclusions justified by the results?

Yes

Is the language acceptable?

Yes

Do you have any ethical concerns with this paper?

No

Have you any concerns about statistical analyses in this paper?

No

Recommendation?

Major revision is needed (please make suggestions in comments)

Comments to the Author(s)

The manuscript entitled "Preparation of novel mature pod-shaped TiO₂ and Ag@TiO₂ nano burst tubes and their photocatalytic activity" reports the fabrication of TiO₂ fibers using electrospinning method and deposited with Ag via impregnation and finally calcination towards the phase formation. H₂C₂O₄ was used to crack down the fibers from tubular shape to near-sheet shape. The objective and experimental design of the work is good. The developed method and obtained results are promising in the field. However, the manuscript lacks in providing substantial evidences and appropriate discussions on the obtained results. Therefore, I recommend for a major revision of the manuscript as per the following comments.

1. What is mechanism behind the cracking of TiO₂ fibers using H₂C₂O₄? Discuss the interaction between H₂C₂O₄ and TBOT
2. Authors should also discuss at which stage the cracking is happening? Is it during the electrospinning or calcination at 550 °C?
3. The discussion on XRD should be improved by discussing the observed intensity changes of the rutile phase with increasing concentration of H₂C₂O₄.
4. It is suggested to discuss whether the morphological changes influenced the crystal structural changes in TiO₂. (the change of tubular morphology to sheet may induce lattice stress in the TiO₂ crystals)
5. It is mentioned that the large surface area of TiO₂(0.3%) is due to the presence of 0.3% H₂C₂O₄. But it is mentioned that H₂C₂O₄ decomposes after the calcination. This should be clarified.
6. The decomposition of H₂C₂O₄ is true with other concentrations also (0.5 and 1%) during the calcination. However, why the decomposition of more H₂C₂O₄ (0.5 and 1%) didn't lead to have the TiO₂ with highly reduced wall thickness and more pores?
7. Provide the pore size distribution curves of the various samples
8. Discuss the mechanism of photodeposition of Ag onto the surface of TiO₂ fibers
9. The discussion on the UV-visible spectra should be improved
10. Estimate the band gap energy of bare TiO₂(0%), TiO₂(0.3%) and Ag-TiO₂ using Tauc plot
11. It is mentioned that "the morphology of mature podshaped TiO₂ NBTs changes as the degradation experiment proceeds". This needs to be supported by experimental evidences such as SEM or TEM. Both for TiO₂ and Ag-TiO₂ samples
12. The photocatalytic efficiency of the system should be studied some colorless pollutants as well (e.g. phenol)
13. The degradation products should be analyzed using TOC
14. Fig. S1 given in the supporting information should be cited (and discussed) in the main manuscript

15. English of the manuscript should be improved

Review form: Reviewer 2

Is the manuscript scientifically sound in its present form?

No

Are the interpretations and conclusions justified by the results?

No

Is the language acceptable?

No

Do you have any ethical concerns with this paper?

No

Have you any concerns about statistical analyses in this paper?

No

Recommendation?

Reject

Comments to the Author(s)

This paper "Preparation of novel mature pod-shaped TiO₂ and Ag@TiO₂ nano burst tubes and their photocatalytic activity" reports the preparation of TiO₂ and Ag@TiO by using electrospinning, which has been used extensively. The idea on the materials design is not clear. The sample preparation was not explained clearly and the products characterization was not enough to conclude the mechanism of the photocatalytic reaction. The reviewer does not recommend publication of this paper.

Comments

1. The introduction (especially the first paragraph) is too general.
2. Why the mass fraction of H₂C₂O₄ is 0%, 0.1%, 0.3%, 0.5% and 1.0% were used? The expected effects of the mass fraction of H₂C₂O₄ on the photocatalyst performances should be clearly explained.
3. Why the samples were calcined at 550 °C, though the products are mixture of rutile and anatase.
4. Experimental section, alpha is missing after K (line 16) and Ka (line 19) should be K alpha.
5. Figure 1 is not clear, especially Figure 1 f (TEM) is not appropriate to discuss the crystallite.
6. P 5, line 18 "Ag@TiO₂ nanocomposites are capable of absorbing RhB molecules" what does this statement mean?
7. In conclusion, the authors claimed "The current work provides a new form of carrier for metal and metal oxide particles, which has a high potential value for future photocatalytic research". The reviewer does not find "new" aspect of the present results and discussion. And there are no experimental data reported for the discussion on the reaction mechanisms.

Review form: Reviewer 3 (Girish Kumar)

Is the manuscript scientifically sound in its present form?

Yes

Are the interpretations and conclusions justified by the results?

Yes

Is the language acceptable?

Yes

Do you have any ethical concerns with this paper?

No

Have you any concerns about statistical analyses in this paper?

No

Recommendation?

Major revision is needed (please make suggestions in comments)

Comments to the Author(s)

Dear authors,

The fabrication of TiO₂ nanotubular structures assisted with polystyrene template and oxalic acid is excellently presented and further decorated with Ag metal via photodeposition to form metal-semiconductor heterostructure. The reasonable characterization and appreciable discussion throughout on the structure-electronic and photocatalytic properties offers some insights into the work. However, morphological evolution of TiO₂ with relevance to varied reaction parameters needs to be discussed for better visibility of the work.

Major comments

- (1) The influence of annealing temperature on the morphology and photocatalytic activity must be discussed.
- (2) The band alignment and Fermi-level equilibration of Ag-TiO₂ indicating the charge carrier dynamics may be presented.

Minor comments

Title: Remove the term 'novel mature' for readability!

Summary

- (1) Remove the term 'mature' in the starting phrase.
- (2) Indicate the optimum content of Ag NPs and its size to attain maximum activity.
- (3) Specify the band gap response of Ag-TiO₂ achieved.

Introduction

- (1) The starting phrase 'TiO₂ has very....international research' may be either removed or condensed as it is too general in its content.

Section 3

- (1) The 'materials characterization' must appear before 'photocatalytic activity measurement' to synchronize with 'Results and discussion' part!

Section 4

- (1) The ripening process leading to tubular structure and further cracking associated with oxalic acid should be properly addressed for more understanding.
- (2) Is it possible to provide TGA/DTA analysis which might provide more information on the decomposition of additives?
- (3) Comment on the amount of Ag metal leaching (if any) from the catalyst surface at the end of the reaction.

Decision letter (RSOS-191019.R0)

24-Jun-2019

Dear Dr Cheng:

Title: Preparation of novel mature pod-shaped TiO₂ and Ag@TiO₂ nano burst tubes and their photocatalytic activity
Manuscript ID: RSOS-191019

The editor assigned to your manuscript has now received comments from reviewers. We would like you to revise your paper in accordance with the referee and Subject Editor suggestions which can be found below (not including confidential reports to the Editor). Please note this decision does not guarantee eventual acceptance.

Please submit your revised paper before 17-Jul-2019. Please note that the revision deadline will expire at 00.00am on this date. If we do not hear from you within this time then it will be assumed that the paper has been withdrawn. In exceptional circumstances, extensions may be possible if agreed with the Editorial Office in advance. We do not allow multiple rounds of revision so we urge you to make every effort to fully address all of the comments at this stage. If deemed necessary by the Editors, your manuscript will be sent back to one or more of the original reviewers for assessment. If the original reviewers are not available we may invite new reviewers.

Please also include the following statements alongside the other end statements. As we cannot publish your manuscript without these end statements included, if you feel that a given heading is not relevant to your paper, please nevertheless include the heading and explicitly state that it is not relevant to your work.

- Acknowledgements

RSC Associate Editor:

Comments to the Author:

Due to conflicting initial reports, a third adjudicative reviewer (Reviewer 3) was invited. On balance, we would like to recommend major revisions.

RSC Subject Editor:

Comments to the Author:

(There are no comments.)

Reviewers' Comments to Author:

Reviewer: 1

Comments to the Author(s)

The manuscript entitled "Preparation of novel mature pod-shaped TiO₂ and Ag@TiO₂ nano burst tubes and their photocatalytic activity" reports the fabrication of TiO₂ fibers using electrospinning method and deposited with Ag via impregnation and finally calcination towards the phase formation. H₂C₂O₄ was used to crack down the fibers from tubular shape to near-sheet shape. The objective and experimental design of the work is good. The developed method and obtained results are promising in the field. However, the manuscript lacks in providing substantial evidences and appropriate discussions on the obtained results. Therefore, I recommend for a major revision of the manuscript as per the following comments.

1. What is mechanism behind the cracking of TiO₂ fibers using H₂C₂O₄? Discuss the interaction between H₂C₂O₄ and TBOT
2. Authors should also discuss at which stage the cracking is happening? Is it during the electrospinning or calcination at 550 °C?

3. The discussion on XRD should be improved by discussing the observed intensity changes of the rutile phase with increasing concentration of H₂C₂O₄.
4. It is suggested to discuss whether the morphological changes influenced the crystal structural changes in TiO₂. (the change of tubular morphology to sheet may induce lattice stress in the TiO₂ crystals)
5. It is mentioned that the large surface area of TiO₂(0.3%) is due to the presence of 0.3% H₂C₂O₄. But it is mentioned that H₂C₂O₄ decomposes after the calcination. This should be clarified.
6. The decomposition of H₂C₂O₄ is true with other concentrations also (0.5 and 1%) during the calcination. However, why the decomposition of more H₂C₂O₄ (0.5 and 1%) didn't lead to have the TiO₂ with highly reduced wall thickness and more pores?
7. Provide the pore size distribution curves of the various samples
8. Discuss the mechanism of photodeposition of Ag onto the surface of TiO₂ fibers
9. The discussion on the UV-visible spectra should be improved
10. Estimate the band gap energy of bare TiO₂(0%), TiO₂(0.3%) and Ag-TiO₂ using Tauc plot
11. It is mentioned that "the morphology of mature podshaped TiO₂ NBTs changes as the degradation experiment proceeds". This needs to be supported by experimental evidences such as SEM or TEM. Both for TiO₂ and Ag-TiO₂ samples
12. The photocatalytic efficiency of the system should be studied some colorless pollutants as well (e.g. phenol)
13. The degradation products should be analyzed using TOC
14. Fig. S1 given in the supporting information should be cited (and discussed) in the main manuscript
15. English of the manuscript should be improved

Reviewer: 2

Comments to the Author(s)

This paper "Preparation of novel mature pod-shaped TiO₂ and Ag@TiO₂ nano burst tubes and their photocatalytic activity" reports the preparation of TiO₂ and Ag@TiO by using electrospinning, which has been used extensively. The idea on the materials design is not clear. The sample preparation was not explained clearly and the products characterization was not enough to conclude the mechanism of the photocatalytic reaction. The reviewer does not recommend publication of this paper.

Comments

1. The introduction (especially the first paragraph) is too general.
2. Why the mass fraction of H₂C₂O₄ is 0%, 0.1%, 0.3%, 0.5% and 1.0% were used? The expected effects of the mass fraction of H₂C₂O₄ on the photocatalyst performances should be clearly explained.
3. Why the samples were calcined at 550 °C, though the products are mixture of rutile and anatase.

4. Experimental section, alpha is missing after K (line 16) and Ka (line 19) should be K alpha.
5. Figure 1 is not clear, especially Figure 1 f (TEM) is not appropriate to discuss the crystallite.
6. P 5, line 18 "Ag@TiO₂ nanocomposites are capable of absorbing RhB molecules" what does this statement mean?
7. In conclusion, the authors claimed "The current work provides a new form of carrier for metal and metal oxide particles, which has a high potential value for future photocatalytic research". The reviewer does not find "new" aspect of the present results and discussion. And there are no experimental data reported for the discussion on the reaction mechanisms.

Reviewer: 3

Comments to the Author(s)

Dear authors,

The fabrication of TiO₂ nanotubular structures assisted with polystyrene template and oxalic acid is excellently presented and further decorated with Ag metal via photodeposition to form metal-semiconductor heterostructure. The reasonable characterization and appreciable discussion throughout on the structure-electronic and photocatalytic properties offers some insights into the work. However, morphological evolution of TiO₂ with relevance to varied reaction parameters needs to be discussed for better visibility of the work.

Major comments

- (1) The influence of annealing temperature on the morphology and photocatalytic activity must be discussed.
- (2) The band alignment and Fermi-level equilibration of Ag-TiO₂ indicating the charge carrier dynamics may be presented.

Minor comments

Title: Remove the term 'novel mature' for readability!

Summary

- (1) Remove the term 'mature' in the starting phrase.
- (2) Indicate the optimum content of Ag NPs and its size to attain maximum activity.
- (3) Specify the band gap response of Ag-TiO₂ achieved.

Introduction

- (1) The starting phrase 'TiO₂ has very....international research' may be either removed or condensed as it is too general in its content.

Section 3

- (1) The 'materials characterization' must appear before 'photocatalytic activity measurement' to synchronize with 'Results and discussion' part!

Section 4

- (1) The ripening process leading to tubular structure and further cracking associated with oxalic acid should be properly addressed for more understanding.
- (2) Is it possible to provide TGA/DTA analysis which might provide more information on the decomposition of additives?
- (3) Comment on the amount of Ag metal leaching (if any) from the catalyst surface at the end of the reaction.

Author's Response to Decision Letter for (RSOS-191019.R0)

See Appendix A.

RSOS-191019.R1 (Revision)

Review form: Reviewer 1

Is the manuscript scientifically sound in its present form?

Yes

Are the interpretations and conclusions justified by the results?

Yes

Is the language acceptable?

Yes

Do you have any ethical concerns with this paper?

No

Have you any concerns about statistical analyses in this paper?

No

Recommendation?

Accept as is

Comments to the Author(s)

Authors have revised the manuscript satisfactorily and it can be accepted for the publication.

Review form: Reviewer 3 (Girish Kumar)

Is the manuscript scientifically sound in its present form?

Yes

Are the interpretations and conclusions justified by the results?

Yes

Is the language acceptable?

Yes

Do you have any ethical concerns with this paper?

No

Have you any concerns about statistical analyses in this paper?

No

Recommendation?

Accept as is

Comments to the Author(s)

Dear authors,

The patience to consider all the comments and point-to-point valid response for each of the queries raised complimented with significant corrections in the article deserves appreciation and revised article deserves to be published.

Decision letter (RSOS-191019.R1)

12-Aug-2019

Dear Dr Cheng:

Title: Preparation of pod-shaped TiO₂ and Ag@TiO₂ nano burst tubes and their photocatalytic activity

Manuscript ID: RSOS-191019.R1

It is a pleasure to accept your manuscript in its current form for publication in Royal Society Open Science. The chemistry content of Royal Society Open Science is published in collaboration with the Royal Society of Chemistry.

RSC Associate Editor:
Comments to the Author:
(There are no comments.)

RSC Subject Editor:
Comments to the Author:
(There are no comments.)

Reviewer(s)' Comments to Author:
Reviewer: 3

Comments to the Author(s)

Dear authors,

The patience to consider all the comments and point-to-point valid response for each of the queries raised complimented with significant corrections in the article deserves appreciation and revised article deserves to be published.

Reviewer: 1

Comments to the Author(s)
Authors have revised the manuscript satisfactorily and it can be accepted for the publication.

Appendix A

Dear Editor and Reviewers:

Many thanks for your letter and the reviewers' comments and suggestions concerning our manuscript entitled *Preparation of pod-shaped TiO₂ and Ag@TiO₂ nano burst tubes and their photocatalytic activity*. Those comments and suggestions are all valuable and helpful for revising and improving our paper, as well as the important guiding significance to our researches. We have studied comments carefully and have made correction which we hope meet with approval. The following is the answers and revisions I have made in response to the reviewers' questions and suggestions on an item by item basis:

Coments:

Editor:

We added Acknowledgements in the manuscript. The content is as follows:

This paper was made better thanks to helpful comments from Shengzhe Zhao, Mingxu Chu, Chaoyue Zhao, Caijiao Ying and Yafeng Yuan.

Reviewer1:

1. What is mechanism behind the cracking of TiO₂ fibers using H₂C₂O₄? Discuss the interaction between H₂C₂O₄ and TBOT.
2. Authors should also discuss at which stage the cracking is happening? Is it during the electrospinning or calcination at 550 °C?
3. The discussion on XRD should be improved by discussing the observed intensity changes of the rutile phase with increasing concentration of H₂C₂O₄.
4. It is suggested to discuss whether the morphological changes influenced the crystal structural changes in TiO₂. (the change of tubular morphology to sheet may induce lattice stress in the TiO₂ crystals).
5. It is mentioned that the large surface area of TiO₂(0.3%) is due to the presence of 0.3% H₂C₂O₄. But it is mentioned that H₂C₂O₄ decomposes after the calcination. This

should be clarified.

6. The decomposition of $\text{H}_2\text{C}_2\text{O}_4$ is true with other concentrations also (0.5 and 1%) during the calcination. However, why the decomposition of more $\text{H}_2\text{C}_2\text{O}_4$ (0.5 and 1%) didn't lead to have the TiO_2 with highly reduced wall thickness and more pores?

7. Provide the pore size distribution curves of the various samples.

8. Discuss the mechanism of photodeposition of Ag onto the surface of TiO_2 fibers.

9. The discussion on the UV-visible spectra should be improved.

10. Estimate the band gap energy of bare TiO_2 (0%), TiO_2 (0.3%) and Ag- TiO_2 using Tauc plot.

11. It is mentioned that "the morphology of mature podshaped TiO_2 NBTs changes as the degradation experiment proceeds". This needs to be supported by experimental evidences such as SEM or TEM. Both for TiO_2 and Ag- TiO_2 samples.

12. The photocatalytic efficiency of the system should be studied some colorless pollutants as well (e.g. phenol).

13. The degradation products should be analyzed using TOC.

14. Fig. S1 given in the supporting information should be cited (and discussed) in the main manuscript.

15. English of the manuscript should be improved.

Reviewer2:

1. The introduction (especially the first paragraph) is too general.

2. Why the mass fraction of $\text{H}_2\text{C}_2\text{O}_4$ is 0%, 0.1%, 0.3%, 0.5% and 1.0% were used? The expected effects of the mass fraction of $\text{H}_2\text{C}_2\text{O}_4$ on the photocatalyst performances should be clearly explained.

3. Why the samples were calcined at 550 °C, though the products are mixture of rutile and anatase.

4. Experimental section, alpha is missing after K (line 16) and K_α (line 19) should be K alpha.

5. Figure 1 is not clear, especially Figure 1 f (TEM) is not appropriate to discuss the crystallite.

6. P 5, line 18 "Ag@ TiO_2 nanocomposites are capable of absorbing RhB molecules" what does this statement mean?

7. In conclusion, the authors claimed "The current work provides a new form of carrier for metal and metal oxide particles, which has a high potential value for future photocatalytic research". The reviewer does not find "new" aspect of the present results

and discussion. And there are no experimental data reported for the discussion on the reaction mechanisms.

Reviewer3:

Major comments

(1) The influence of annealing temperature on the morphology and photocatalytic activity must be discussed.

(2) The band alignment and Fermi-level equilibration of Ag-TiO₂ indicating the charge carrier dynamics may be presented.

Minor comments

Title: Remove the term 'novel mature' for readability!

Summary

(1) Remove the term 'mature' in the starting phrase.

(2) Indicate the optimum content of Ag NPs and its size to attain maximum activity.

(3) Specify the band gap response of Ag-TiO₂ achieved.

Introduction

(1) The starting phrase 'TiO₂ has very....international research' may be either removed or condensed as it is too general in its content.

Section 3

(1) The 'materials characterization' must appear before 'photocatalytic activity measurement' to synchronize with 'Results and discussion' part!

Section 4

(1) The ripening process leading to tubular structure and further cracking associated with oxalic acid should be properly addressed for more understanding.

(2) Is it possible to provide TGA/DTA analysis which might provide more information on the decomposition of additives?

(3) Comment on the amount of Ag metal leaching (if any) from the catalyst surface at the end of the reaction.

Response to reviewer1:

1. What is mechanism behind the cracking of TiO₂ fibers using H₂C₂O₄? Discuss the interaction between H₂C₂O₄ and TBOT.

(1) What is mechanism behind the cracking of TiO₂ fibers using H₂C₂O₄?

The mechanism of H₂C₂O₄ cracking TiO₂ fiber is very simple. First, H₂C₂O₄ is added to the electrospinning solution. Polystyrene (PS) nanofibers with H₂C₂O₄ were prepared by electrospinning, and PS nanofibers were used as a sacrificial template for the preparation of TiO₂ nanotubes. During the heating and calcination process, the PS nanofibers and H₂C₂O₄ will completely decompose. In particular, H₂C₂O₄ releases gas during the decomposition process. The decomposition temperature of H₂C₂O₄ (250 °C) is lower than the decomposition temperature of PS, so under the same calcination conditions, H₂C₂O₄ will preferentially decompose. And the presence of a large amount of gas causes the PS fiber structure to be looser, thereby accelerating the decomposition of PS. At the same time, 350~450 °C is the key period for the fixed shape of TiO₂ nanotubes. When TiO₂ nanotubes are subjected to a large amount of gas (H₂O and CO₂ produced by decomposition of PS) during this critical period, the TiO₂ nanotubes will be cracked to form a pod-shaped TiO₂ NBTs, and the surface of the TiO₂ NBTs will be have a lot of gas overflows the hole.

(2) Discuss the interaction between H₂C₂O₄ and TBOT.

There is no interaction between H₂C₂O₄ and TBOT. Since the complete decomposition temperature of H₂C₂O₄ is 250 °C, tetrabutyl titanate (TBOT) is subjected to aerobic calcination to form TiO₂ at 350 °C. Therefore, H₂C₂O₄ is preferentially decomposed into H₂O and CO₂ gases. Because 350~450 °C is the key period for TiO₂ NBTs shaping. Therefore, the presence of H₂C₂O₄ indirectly leads to a thinner and more brittle wall of

the TiO₂ nanotubes, and cracking occurs during the heating and calcination process to form a pod-shaped TiO₂ NBTs.

2. Authors should also discuss at which stage the cracking is happening? Is it during the electrospinning or calcination at 550 °C?

The cracking of TiO₂ nanotubes is carried out during the calcination process at a temperature of 550 °C. First, we added H₂C₂O₄ to the electrospinning solution. PS nanofibers with H₂C₂O₄ were prepared by electrospinning, and PS nanofibers were used as a sacrificial template for the preparation of TiO₂ nanotubes. Secondly, in the aerobic calcination process, the PS nanofibers and H₂C₂O₄ will completely decompose, leaving only the TiO₂ NBTs. Therefore, the crack generated by TiO₂ is during the calcination at 550 °C.

3. The discussion on XRD should be improved by discussing the observed intensity changes of the rutile phase with increasing concentration of H₂C₂O₄.

Fig. 1 The XRD patterns of the five samples

It can be seen from Fig. 1 that the TiO₂ crystal forms of TiO₂ NBTs prepared by adding H₂C₂O₄ mass fractions of 0%, 0.1%, 0.3%, 0.5% and 1.0% are all identical.

The main diffraction peaks of TiO₂ at 25.28°, 37.80°, 48.05°, 53.89°, 55.06 and

62.69° can be directed to the (101), (004), (200), (105), (211) and (204) directions.

Characterizing the anatase structure of TiO₂ (JCPDS 21-1272). The main diffraction peaks of TiO₂ at 27.45° and 36.09° point in the (110) and (101) directions and are characterized by rutile TiO₂ (JCPDS 21-1276). The peak of the XRD pattern corresponding to the TiO₂ anatase structure is very sharp and has a plurality of corresponding peaks, and thus these peaks serve as main product peaks. The peak of the XRD pattern of the TiO₂ rutile structure is very small and can only be used as a by-product peak. As the concentration of H₂C₂O₄ increased, the strength of the rutile phase did not change. Therefore, we have not improved the discussion of sample XRD.

4. It is suggested to discuss whether the morphological changes influenced the crystal structural changes in TiO₂. (the change of tubular morphology to sheet may induce lattice stress in the TiO₂ crystals).

The morphological change of the sample does not affect the crystal structure change of TiO₂. Since it can be observed from above Fig. 1, all of the five samples calcined at 550 °C have the same crystal type. The main diffraction peaks of TiO₂ at 25.28°, 37.80°, 48.05°, 53.89°, 55.06 and 62.69° can be directed to the (101), (004), (200), (105), (211) and (204) directions. Characterizing the anatase structure of TiO₂ (JCPDS 21-1272). The main diffraction peaks of TiO₂ at 27.45° and 36.09° point in the (110) and (101) directions and are characterized by rutile TiO₂ (JCPDS 21-1276). The peak of the XRD pattern corresponding to the TiO₂ anatase structure is very sharp and has a plurality of corresponding peaks, and thus these peaks serve as main product peaks. The peak corresponding to the TiO₂ rutile structure serves as a by-product peak. The change of

the TiO_2 from the tubular shape to the sheet does not affect the crystal structure change of TiO_2 .

5. It is mentioned that the large surface area of $\text{TiO}_2(0.3\%)$ is due to the presence of $0.3\% \text{H}_2\text{C}_2\text{O}_4$. But it is mentioned that $\text{H}_2\text{C}_2\text{O}_4$ decomposes after the calcination. This should be clarified.

Fig.2 TG-DTA analysis of (a) $\text{H}_2\text{C}_2\text{O}_4$ and (b) $\text{TiO}_2(0.3\%)$ precursor fiber soaked in TBOT.

As can be seen from Fig. 2(a), the complete decomposition temperature of $\text{H}_2\text{C}_2\text{O}_4$ was 250°C . Fig. 2(b) shows the picture of weight loss when pyrolysis of $\text{TiO}_2(0.3\%)$ precursor fiber soaked with TBOT. There was no significant change in the weight loss of the sample before 250°C . The mass fraction of $\text{H}_2\text{C}_2\text{O}_4$ in the $\text{TiO}_2(0.3\%)$ precursor fiber soaked in TBOT was only 0.3% . Therefore, during the heating and calcination process, the PS nanofibers and $\text{H}_2\text{C}_2\text{O}_4$ are completely decomposed. In particular, $\text{H}_2\text{C}_2\text{O}_4$ releases gas during the decomposition process. The decomposition temperature of $\text{H}_2\text{C}_2\text{O}_4$ (250°C) is lower than the decomposition temperature of PS, so under the same calcination conditions, $\text{H}_2\text{C}_2\text{O}_4$ will preferentially decompose. And the presence of a large amount of gas causes the PS fiber structure to be looser, thereby accelerating the decomposition of PS. At the same time, $350\sim 450^\circ\text{C}$ is the key period for the fixed shape of TiO_2 NBTs. When TiO_2 NBTs are impacted by a large amount of gas (the gas

source is H₂O and CO₂ generated by PS decomposition) in this critical period, TiO₂ NBTs will crack and form pod-shaped TiO₂ NBTs, and there will be a lot of gas overflow holes on the surface of TiO₂ NBTs. Therefore, the specific surface area of TiO₂ NBTs is increased.

6. The decomposition of H₂C₂O₄ is true with other concentrations also (0.5 and 1%) during the calcination. However, why the decomposition of more H₂C₂O₄ (0.5 and 1%) didn't lead to have the TiO₂ with highly reduced wall thickness and more pores?

Fig. 3 SEM images of five samples: (a)-(e) were sampled at TiO₂(0%), TiO₂(0.1%), TiO₂(0.3%), TiO₂(0.5%) and TiO₂(1.0%), respectively (The illustration in Fig. 1(c) shows the sample crack width statistics), (f) TEM image and (g) HRTEM image of sample TiO₂(0.3%), (h) the XRD patterns of the five samples.

In fact, more decomposition of H₂C₂O₄ (0.5 and 1%) leads to thinning of the wall of the TiO₂ NBTs. It can be clearly observed from Fig. 3 (d) and (e) that the walls of the TiO₂(0.5%) and TiO₂(1.0%) NBTs are thin. In particular, TiO₂ (1.0%) of Fig. 3(e) has become a gauze-like nanobelt.

Fig. 4 Nitrogen sorption-desorption isotherms of the samples.

As shown in Fig. 4, the specific surface area of the samples were 28.16, 65.35, 67.55, 43.42 and 51.89 $\text{m}^2 \text{g}^{-1}$. Sample $\text{TiO}_2(0.3\%)$ have large specific surface area. All samples exhibited the same hysteresis loop in the range of 0.3-1.0 P/P_0 , indicating a similar pore structure between samples. The specific surface area of $\text{TiO}_2(0.5\%)$ and $\text{TiO}_2(1.0\%)$ became small by the BET test on the sample. Because the normal TiO_2 NBTs ($\text{TiO}_2(0.3\%)$) is well-defined, the tubes and tubes will maintain a suitable distance. However, the degree of cracking of the TiO_2 NBTs is increased, which directly causes an increase in the overlap between the TiO_2 tubes and the tubes, and multiple nano burst tubes or a plurality of nanobelts are stuck together, thereby affecting the BET test result of the sample, resulting in The specific surface area of the sample becomes smaller and the pores are less.

7. Provide the pore size distribution curves of the various samples.

Fig. 5 BJH pore size distribution of five samples.

Fig. 5 shows the BJH pore size distribution of five samples, and the porous structure of the sample was investigated. The total pore volumes of $\text{TiO}_2(0\%)$, $\text{TiO}_2(0.1\%)$, $\text{TiO}_2(0.3\%)$, $\text{TiO}_2(0.5\%)$ and $\text{TiO}_2(1.0\%)$ were 0.14, 0.36, 0.52, 0.25 and 0.23 $\text{cm}^3 \text{g}^{-1}$, respectively. The average pore diameters were 14.05, 16.12, 25.39, 13.74 and 14.57 nm, respectively. The $\text{TiO}_2(0.3\%)$ NBTs (Fig. 5c) has the largest total pore volume and average pore size while having the largest specific surface area. It also showed excellent performance in photocatalytic performance testing.

8. Discuss the mechanism of photodeposition of Ag onto the surface of TiO_2 fibers.

The deposition of AgNPs on the surface of the pod-shaped TiO_2 NBTs is very simple. Mainly by the irradiation of ultraviolet light, glucose is used to reduce silver nitrate to prepare AgNPs. It is particularly important that the silver nitrate aqueous glucose solution be mixed with the pod-shaped TiO_2 NBTs before mixing the AgNPs. The reason is that the AgNPs produced by the reduction of ultraviolet light can be effectively

grown in the pod-shaped TiO₂ NBTs.

9. The discussion on the UV-visible spectra should be improved.

Fig. 6 UV-vis diffuse reflectance spectra of samples TiO₂(0.3%) and (0.05M)Ag@TiO₂(0.3%).

Fig. 6 shows the UV-visible diffuse reflectance spectra of samples TiO₂(0.3%) and (0.05M)Ag@TiO₂(0.3%). Use the intercept method to easily determine the semiconductor forbidden band width. The tangent axes of the TiO₂(0.3%) and (0.05M)Ag@TiO₂(0.3%) were 424.66 nm and 454.21 nm, respectively, and the forbidden band widths (E_g) of the two samples were 2.92 eV and 2.73 eV, respectively. Therefore, the absorption light range of TiO₂(0.3%) loaded with silver nanoparticles moves in the visible light direction, which improves the light utilization efficiency of the sample.

10. Estimate the band gap energy of bare TiO₂(0%), TiO₂(0.3%) and Ag-TiO₂ using Tauc plot.

Fig. 7 Tauc plot of $\text{TiO}_2(0.3\%)$ and $(0.05\text{M})\text{Ag}@TiO_2(0.3\%)$

We have determined the band gap energy of bare $\text{TiO}_2(0\%)$ in our work, but bare $\text{TiO}_2(0\%)$ and Tauc plot of $\text{TiO}_2(0.3\%)$ completely coincide, so only have the Tauc plot (Fig. 7) of $\text{TiO}_2(0.3\%)$ and $(0.05\text{M})\text{Ag}@TiO_2(0.3\%)$. This phenomenon also fully shows that the addition of $\text{H}_2\text{C}_2\text{O}_4$ did not change the band gap energy of the TiO_2 NBTs. According to the Tauc plot method, the band gap energy of $\text{TiO}_2(0.3\%)$ is 2.92 eV and the $(0.05\text{M})\text{Ag}@TiO_2(0.3\%)$ band gap energy of deposited AgNPs is 2.73 eV. Therefore, the band gap energy is reduced after the sample is loaded with the AgNPs, and the utilization efficiency of light is improved.

11. It is mentioned that "the morphology of mature podshaped TiO_2 NBTs changes as the degradation experiment proceeds". This needs to be supported by experimental evidences such as SEM or TEM. Both for TiO_2 and Ag- TiO_2 samples.

Fig. 8 SEM image of pod-shaped $\text{TiO}_2(0.3\%)$ (a-b) and $(0.05\text{M})\text{Ag}@TiO_2(0.3\%)$ (c-d) NBTs before and after the catalytic experiment.

As shown in the SEM image of Fig. 8, the morphology of the pod-shaped $\text{TiO}_2(0.3\%)$ NBTs (a) and $(0.05\text{M})\text{Ag}@\text{TiO}_2(0.3\%)$ NBTs (c) was clear before the catalytic experiment. The nanotubes is evenly connected. However, when the reaction was completed, the recovered pod-shaped $\text{TiO}_2(0.3\%)$ NBTs (b) and $(0.05\text{M})\text{Ag}@\text{TiO}_2(0.3\%)$ NBTs (d) were subjected to scanning electron microscopy (SEM). The results showed that the structure of the catalyst after the reaction changed. The sample is transformed from a uniformly cracked nano burst tubes into a broken nano burst tubes. Therefore, the morphology of the of pod-shaped $\text{TiO}_2(0.3\%)$ NBTs changes with the degradation experiment.

12. The photocatalytic efficiency of the system should be studied some colorless pollutants as well (e.g. phenol).

Reviewers have suggested that it is very meaningful to degrade colorless pollutants. We are very sorry, there is no degradation of colorless contaminants such as phenol in manuscript. Because our work is mainly to propose that we have successfully prepared a shape-controllable pod-shaped TiO_2 NBTs, and compared the catalytic performance of TiO_2 NBTs and ordinary TiO_2 nanotubes. Therefore, we only used a simple colored RhB solution to discuss the catalytic performance of the prepared samples. We plan to complete the catalytic degradation of colorless pollutants (such as phenol) by pod-shaped TiO_2 NBTs in the next step. I look forward to further communication with you.

13. The degradation products should be analyzed using TOC.

Thanks to the reviewer for his suggestions. When degrading organic pollutants, we can determine the degradation of pollutants by measuring the absorbance of the solution and the total organic carbon (TOC). However, for colored organic pollutants (RhB), it

is simpler and more convenient to directly measure the absorbance of the solution. Therefore, only the content of the solution absorbance is measured in our manuscript. The reviewer's suggestion to determine the TOC in the solution is very meaningful. We plan to degrade the colorless pollutants in the next step and determine the contents of the TOC solution to enrich the experimental content.

14. Fig. S1 given in the supporting information should be cited (and discussed) in the main manuscript.

Thanks for your suggestion, we decided to add the content of the support information to the body of the manuscript for your reading and understanding. We added the $\text{H}_2\text{C}_2\text{O}_4$ decomposition curve to enrich the experimental content. Add content in a red font in the manuscript.

Fig.9 TG-DTA analysis of (a) $\text{H}_2\text{C}_2\text{O}_4$ and (b) $\text{TiO}_2(0.3\%)$ precursor fiber soaked in TBOT.

TG-DTA analysis of pure $\text{H}_2\text{C}_2\text{O}_4$ and PS/TBOT composite fiber samples with $\text{H}_2\text{C}_2\text{O}_4$ content of 0.3%, the result was shown in Fig. 9. As can be seen from Fig. 9(a), the complete decomposition temperature of pure $\text{H}_2\text{C}_2\text{O}_4$ is 250°C and $\text{H}_2\text{C}_2\text{O}_4$ has been completely decomposed when PS has not been decomposed. Fig. 9(b) shows that the thermal decomposition process of the sample was divided into three stages. The first stage occurred between $70^\circ\text{C}\sim 350^\circ\text{C}$. The sample lost water and $\text{H}_2\text{C}_2\text{O}_4$ decomposed

to release CO₂ gas. The second stage is 350~450°C, which represents the decomposition of PS organic components and some of the organics produced by TBOT hydrolysis. The third stage is 450°C~700°C, mainly including PS main chain degradation and amorphous TiO₂ to anatase phase two processes.

15. English of the manuscript should be improved.

Thanks to the reviewer's suggestion, we used a professional language editing service to improve the English of the manuscript.

ELSEVIER
Language Editing Services
Registered Office:
Elsevier Ltd
The Boulevard, Langford Lane,
Kidlington, OX5 1GB, UK.
Registration No. 331566771

To whom it may concern

The paper "Preparation of pod-shaped TiO₂ and Ag@TiO₂ nano burst tubes and their photocatalytic activity" by Shang Wang was edited by Elsevier Language Editing Services.

Kind regards,

Elsevier Webshop Support

Response to reviewer2:

1. The introduction (especially the first paragraph) is too general.

‘TiO₂ has very....international research’ is indeed too general, and we decided to concentrate and modify this part. **The modified content is indicated in red font in the body of the manuscript.** The changes are as follows:

Titanium dioxide (TiO₂) a multifunctional semiconductor metal oxide, which has

attracted an extensive range of research attention because of their unique optical, electronic, and antibacterial properties. In practical applications, TiO₂ is not toxic, has super-hydrophilicity, and can completely contact with food[1] and does not affect human health. TiO₂ shows good development prospects in many fields. At present, TiO₂ has been widely used as a multifunctional material in the fields of solar cells[2], sensors[3], ceramics[4, 5], especially photocatalysts[6-8] in many environmental pollution control programs. Photocatalytic technology based on nano-TiO₂ materials provides a cheap, non-toxic, energy-efficient, and highly efficient method for degrading organic pollutants in air and water[9-12]. As an indirect bandgap semiconductor material, TiO₂ has typical semiconductor energy band characteristics. The energy band consists of a valence band (VB) filled with electron orbits, an empty orbital conduction band (CB) without electrons, and a band gap (E_g) between the valence band and the conduction band. When TiO₂ is not excited, electrons in the valence band do not automatically transition to the conduction band. Only when the energy excited by the photons is greater than E_g, the electrons in the valence band absorb the energy of the photon transition into the conduction band, and holes are generated in the valence band, "electron-hole pairs" (e⁻-h⁺) are formed[13-16]. As the research groups continue to explore, the morphological structure of TiO₂ is also constantly changing. Nanorods, nanotubes, nanoflowers, nanoparticles and nanofibers have been studied and prepared. Although notable advances have been made, but the high recombination rate of the photogenerated electron/hole pairs and the low utilization rate of ultraviolet hinders its further application in industry.

2. Why the mass fraction of $\text{H}_2\text{C}_2\text{O}_4$ is 0%, 0.1%, 0.3%, 0.5% and 1.0% were used? The expected effects of the mass fraction of $\text{H}_2\text{C}_2\text{O}_4$ on the photocatalyst performances should be clearly explained.

(1) First, the use of $\text{H}_2\text{C}_2\text{O}_4$ in experiments to cause changes in the morphology of TiO_2 nanotubes is an unexpected finding. We do not know that the addition of $\text{H}_2\text{C}_2\text{O}_4$ will affect the morphology of TiO_2 nanotubes. Moreover, we do not know how much $\text{H}_2\text{C}_2\text{O}_4$ will be added to produce this pod-shaped TiO_2 NBTs. Therefore, the five $\text{H}_2\text{C}_2\text{O}_4$ mass fractions of 0%, 0.1%, 0.3%, 0.5% and 1.0% were used to determine whether the amount of $\text{H}_2\text{C}_2\text{O}_4$ added would affect the morphology of TiO_2 nanotubes.

(2) The expected result of our experiment is that as the amount of $\text{H}_2\text{C}_2\text{O}_4$ added increases, the catalytic performance of the sample after calcination will be better. Because $\text{H}_2\text{C}_2\text{O}_4$ can be completely decomposed at 250 °C, H_2O and CO_2 gases are released. When the amount of $\text{H}_2\text{C}_2\text{O}_4$ added is increased, it means that more gas is generated, resulting in a looser structure of the PS fiber, which leads to rapid decomposition of PS. 350~450 °C is the key period for the formation of TiO_2 nanotubes. When TiO_2 nanotubes are subjected to a large amount of gas (H_2O and CO_2 produced by decomposition of PS) during this critical period, TiO_2 nanotubes are cracked, and there are many gas overflow holes on the surface of TiO_2 nanotubes. At the same time, the specific surface area and photocatalytic ability of the sample are improved.

In fact, the results of the degradation experiment are not exactly the same as our expected results. The specific result is that the specific surface area and catalytic ability of the sample increase when the mass fraction of $\text{H}_2\text{C}_2\text{O}_4$ is 0%, 0.1% and 0.3%. However, when the $\text{H}_2\text{C}_2\text{O}_4$ mass fraction is 0.5% and 1.0%, the specific surface area

decreases and the catalytic ability also decreases a lot. This shows that only the appropriate mass fraction (0.3%) of $\text{H}_2\text{C}_2\text{O}_4$ can increase the specific surface area and catalytic ability of TiO_2 nanotubes.

3. Why the samples were calcined at 550 °C, though the products are mixture of rutile and anatase.

Fig. 1 The XRD patterns of the five samples.

It can be observed from Fig. 1 that all of the five samples calcined at 550 °C have the same crystal type. The main diffraction peaks of TiO_2 at 25.28°, 37.80°, 48.05°, 53.89°, 55.06 and 62.69° can be directed to the (101), (004), (200), (105), (211) and (204) directions. Characterizing the anatase structure of TiO_2 (JCPDS 21-1272). The main diffraction peaks of TiO_2 at 27.45° and 36.09° point in the (110) and (101) directions and are characterized by rutile TiO_2 (JCPDS 21-1276). The peak of the XRD pattern corresponding to the TiO_2 anatase structure is very sharp and has a plurality of corresponding peaks, and thus these peaks serve as main product peaks. The peak of the XRD pattern of the TiO_2 rutile structure is very small and can only be used as a by-product peak. Therefore, the product is a mixture of rutile and anatase, and anatase TiO_2 is the main product.

4. Experimental section, alpha is missing after K (line 16) and Ka (line 19) should be K alpha.

We are very sorry for the confusion for editors and reviewers caused by our negligence in the writing process. We have already checked the manuscript and corrected the above mistakes.

The crystal structure properties of pod-shaped TiO_2 NBTs and Ag@TiO_2 NBTs were

studied by X-ray diffractometry (XRD, XRD-7000, Shimadzu) with Cu **K alpha** ($\lambda = 0.15418$ nm) as radiation source and the scanning range from 20° to 80° . Measure the specific surface area of the sample by Specific surface & pore size analysis instrument (3H-2000PS1, BeiShiDe Instrument). X-ray photoelectron spectroscopy (XPS) was performed on a VG ESCALAB LKII instrument with Mg-**K alpha**-ADES ($h\nu = 1253.6$ eV) source at a residual gas pressure of below 1028 Pa.

5. Figure 1 is not clear, especially Figure 1 f (TEM) is not appropriate to discuss the crystallite.

Fig. 2 SEM images of five samples: (a)-(e) were sampled at $\text{TiO}_2(0\%)$, $\text{TiO}_2(0.1\%)$, $\text{TiO}_2(0.3\%)$, $\text{TiO}_2(0.5\%)$ and $\text{TiO}_2(1.0\%)$, respectively (The illustration in Fig. 1(c) shows the sample crack width statistics), (f) TEM image and (g) HRTEM image of sample $\text{TiO}_2(0.3\%)$, (h) the XRD patterns of the five samples.

Sorry, the Figure 1 we gave is not clear enough, especially the (f) in Figure 1 is not suitable for discussion of microcrystals. Therefore, we have reworked Figure 1. In the response letter, we renamed Figure 1 to Fig. 2. In Fig. 2(a-e) respectively describe TiO_2 of five different morphologies. The reason for the formation of different morphologies is the addition of $\text{H}_2\text{C}_2\text{O}_4$ during the electrospinning process. It can be clearly seen from Fig. 2 (a-e) that as the amount of $\text{H}_2\text{C}_2\text{O}_4$ added increases, the degree of cracking of the TiO_2 nanotubes also changes greatly. It can be obtained by observing the Fig. 2(c) that

the surface morphology of $\text{TiO}_2(0.3\%)$ is the best, the cracking is uniform, and the crack is maintained at 200 nm to 400 nm. It also has good catalytic ability in the subsequent photocatalytic performance test. Therefore, a sample of $\text{TiO}_2(0.3\%)$ was selected for TEM (Fig. 2(f)). It can be clearly seen in Fig. 2(f) that there are small white spots on the surface of the sample. In fact, these are the pores on the surface of the sample, which increase the specific surface area of the sample. Fig. 2(g) is an HRTEM image of $\text{TiO}_2(0.3\%)$, and the lattice spacing d of the sample was measured to be 0.351 nm.

The description of the added portion of Figure 1 in the manuscript is indicated in red.

6. P 5, line 18 “ Ag@TiO_2 nanocomposites are capable of absorbing RhB molecules” what does this statement mean?

Fig. 3 Nitrogen sorption-desorption isotherms of the samples

Sorry, our language is not rigorous enough to cause problems for reviewers. The meaning of “ Ag@TiO_2 nanocomposites are capable of absorbing RhB molecules” is that the Ag@TiO_2 nanocomposites we prepared have the ability to adsorb dye molecules. It can be seen from Fig. 3 that the prepared pod-shaped TiO_2 NBTs has the same type of hysteresis loop as ordinary TiO_2 nanotubes, which proves that there is a similar pore structure between these samples. Moreover, the specific surface area of the pod-shaped TiO_2 NBTs increases, which also indicates that the sample with $\text{H}_2\text{C}_2\text{O}_4$

added has a stronger adsorption capacity. At the same time, Ag@TiO₂ NBTs are Ag@TiO₂ NBTs prepared by photoreduction based on pod-shaped TiO₂ NBTs. Therefore, Ag@TiO₂ NBTs also have a larger specific surface area and can adsorb more RhB molecules.

7. In conclusion, the authors claimed “The current work provides a new form of carrier for metal and metal oxide particles, which has a high potential value for future photocatalytic research”. The reviewer does not find “new” aspect of the present results and discussion. And there are no experimental data reported for the discussion on the reaction mechanisms.

Dear reviewer, "new" in "The current work provides a new form of carrier for metal and metal oxide particles, which has a high potential value for future photocatalytic research" means that we successfully prepared the pod-shaped TiO₂ NBTs, and the degree of cracking of this tube is completely controlled by humans. This is a new form that has not appeared in the research of TiO₂ nanotubes. Moreover, because of the addition of oxalic acid, the pod-shaped TiO₂ NBTs has a larger specific surface area, which is more favorable for the catalyst to adsorb dye molecules. It can be seen from the catalytic degradation experiments that the degradation performance of the pod-shaped TiO₂ NBTs is better than that of the ordinary TiO₂ nanotubes (the same illumination time, the degradation efficiency of the pod-shaped TiO₂ NBTs is 91%, ordinary TiO₂ nanotubes is 68%). These experimental data and results show that the work we do is valuable. At the same time, in the next study, the pod-shaped TiO₂ NBTs can be used as a base material, and different metals and metal oxide particles are loaded to improve the ability of TiO₂ to catalytically degrade organic pollutants. Catalytic research has positive implications. Therefore, it can be said that our work provides a

new form of carrier for metal and metal oxide particles.

Response to reviewer3:

Major comments

(1) The influence of annealing temperature on the morphology and photocatalytic activity must be discussed.

In fact, during the experiment we also explored the effects of different calcination temperatures on the catalytic ability of the pod-shaped TiO₂ NBTs. However, when the calcination temperature was 350 °C and 750 °C, the sample had almost no catalytic ability. Therefore, the experimental contents when the calcination temperatures are 350 °C and 750 °C are not described in the manuscript. Here are our additional content:

Fig. 1 SEM image of calcination temperature of 350 °C (a), 550 °C (b), 750 °C (c) and photocatalytic activity of photocatalyst for photodegradation of RhB under ultraviolet light irradiation (d).

When the calcination temperature is 350 °C, Fig. 1(a) shows that only a small portion of the TiO₂ nanotubes is cracked. The reason is that H₂O and CO₂ produced by the decomposition of H₂C₂O₄ and PS are very slow at low temperature calcination. Therefore, the gas gently escapes from the surface of the TiO₂ NBTs, resulting in TiO₂

not being well cracked. However, when the temperature rises to 550 °C (Fig. 1(b)) and 750 °C (Fig. 1(c)), H₂C₂O₄ and PS will rapidly decompose the H₂O and CO₂ gases generated. The impact of a large amount of gas will cause the surface of the TiO₂ nanotube to crack and form a pod-shaped TiO₂ NBTs. Therefore, there is almost no difference in the SEM pattern formed by calcination at 550 °C and 750 °C. The catalytic degradation experiments were carried out on three samples. The experimental results in Fig. 1(d) show that the degradation efficiency of the TiO₂ NBTs calcined at 550 °C is 94.0%, while the degradation efficiency at 550 °C and 750 °C is 30.7. % and 20.1%, so only the catalytic performance of the 550 °C sample was analyzed in the manuscript.

(2) The band alignment and Fermi-level equilibration of Ag-TiO₂ indicating the charge carrier dynamics may be presented.

Thanks to the reviewer for his suggestions. We give the band alignment of the Ag@TiO₂ and the Fermi level balance to show the charge carrier dynamics, which can improve the academic level of the article. The following is the modification:

Fig. 2 (a) Calculated band edge (VB, CB) position of TiO₂(0.3%) and (0.05M)Ag@TiO₂(0.3%) NBTs; (b) Schematic diagram of charge separation and transfer of pod-shaped Ag@TiO₂ NBTs under visible light irradiation.

Fig. 2(a) shows the band edge of the samples TiO₂(0.3%) and (0.05M)Ag@TiO₂(0.3%).

The empirical formula for the edge calculation of semiconductors is:

$$E_{CB} = X - E_e - 0.5Eg$$

$$E_{VB} = E_{CB} + Eg$$

E_{CB} is the CB edge potential, and X is the electronegativity of the semiconductor, which is the arithmetic mean of the electronegativity of constituent atoms and first ionization energy. E_e is the energy of free electrons on the hydrogen scale (approximately 4.5 eV), and Eg is the band gap energy of the semiconductor (According to Tauc plot of the samples, the Eg values of $TiO_2(0.3\%)$ and $(0.05M)Ag@TiO_2(0.3\%)$ are 2.92 eV and 2.73 eV.).

The band edge positions (VB, CB) of $TiO_2(0.3\%)$ were 2.77 eV and -0.15 eV; the band edge positions of $(0.05M)Ag@TiO_2(0.3\%)$ were 2.67 eV and -0.06 eV. According to the band edge position (VB, CB) of $TiO_2(0.3\%)$ and $(0.05M)Ag@TiO_2(0.3\%)$, it can be known that when the AgNPs are deposited on the surface of TiO_2 , the forbidden band width and band edge of the sample are effectively reduced. Therefore, $(0.05M)Ag@TiO_2(0.3\%)$ has a good catalytic effect.

Fig. 2(b) supplements the Fermi level of $(0.05M)Ag@TiO_2(0.3\%)$ to make the charge separation and transfer schematics clearer. The Ag valence state of AgNPs prepared by photoreduction method is 0. When Ag is irradiated by visible light, Ag can be excited, photoelectrons migrate from Ag to TiO_2 , and Ag becomes Ag^+ . At the same time, due to the large work function of Ag, a Schottky barrier is formed at the $Ag@TiO_2$ interface. Thus, AgNPs absorb visible light and produce photogenerated electron/hole pairs, and the generated electrons are transferred to the CB of the anatase phase. A detailed explanation can be obtained from Fig. 11 of the manuscript.

Minor comments

Title: Remove the term ‘novel mature’ for readability!

Thanks for the reviewer's suggestion. The title was modified to ‘Preparation of pod-shaped TiO₂ and Ag@TiO₂ nano burst tubes and their photocatalytic activity’. The title expressed in red font.

Summary

(1) Remove the term ‘mature’ in the starting phrase.

Thanks for the reviewer's suggestion. We removed the ‘mature’ from the summary section. We have put the revised content into the manuscript. In order to maintain the consistency of phrases in the manuscript, we deleted all the "mature" in the body so as to facilitate reviewers to read.

(2) Indicate the optimum content of AgNPs and its size to attain maximum activity.

Fig. 3 The EDX spectra of (a) TiO₂(0.3%), (b) (0.01M)Ag@TiO₂(0.3%), (c) (0.05M)Ag@TiO₂(0.3%) and (d) (0.10M)Ag@TiO₂(0.3%).

To further understand the chemical composition of the obtained Ag@TiO₂ NBTs, energy dispersive X-ray (EDX) analysis was carried out. From the Fig. 3 we can

observe the EDX spectrum of the TiO₂ with different concentration of AgNO₃, where (a) TiO₂(0.3%), (b) (0.01M)Ag@TiO₂(0.3%), (c) (0.05M)Ag@TiO₂(0.3%) and (d) (0.10M)Ag@TiO₂(0.3%). The peaks of the Ti, O and Ag are clearly observable and the atom percent was 74:24:0; 76:23:0.08; 74:25:0.12 and 70:24:0.22, respectively. Result of increasing concentration of AgNO₃, the amount of deposited AgNPs on the surface of the Ag@TiO₂ NBTs increased. This due to the higher concentration of AgNO₃ solution causes more AgNPs to accumulate on the surface of the TiO₂ NBTs, which leads to an increase in particle size, resulting in a decrease in the specific surface area of the particles and the surface active sites. Therefore, when the atomic percentage of the peaks of Ti, O and Ag is 74:25:0.12, the sample has the strongest catalytic ability.

(3) Specify the band gap response of Ag-TiO₂ achieved.

Fig. 4 UV-vis diffuse reflectance spectra (a) and Tauc plot (b) of samples TiO₂(0.3%) and (0.05M)Ag@TiO₂(0.3%).

There are two ways to get the semiconductor bandwidth from the UV-vis DRS spectrum. One is to use the intercept method to easily determine the semiconductor forbidden band width. The basic principle is that the band edge wavelength of the semiconductor (also called the absorption threshold, λ_g) is determined by the forbidden band width E_g , and there is a quantitative relationship between $E_g(\text{eV})=1240/\lambda_g$ (nm). Therefore, E_g

can be obtained by obtaining λ_g . The other is to determine the forbidden band width of the semiconductor by the Tauc plot method. Mainly based on the formula proposed by Tauc, Davis and Mott et al:

$$(\alpha h\nu)^{1/n} = A(h\nu - E_g)$$

Where α is the absorption coefficient, h is the Planck constant, ν is the frequency, A is a constant, E_g is the semiconductor forbidden band width, and the index n is directly related to the semiconductor type (anatase TiO_2 is an indirect bandgap semiconductor, n is 2).

Fig. 4(a) uses the intercept method to determine the λ_g of the sample $\text{TiO}_2(0.3\%)$ and $(0.05\text{M})\text{Ag}@\text{TiO}_2(0.3\%)$ of 424.66 nm and 454.21 nm, respectively. Therefore, the forbidden band width (E_g) of the two samples was 2.92 eV and 2.73 eV, respectively.

Fig. 4(b) uses the Tauc plot method to determine the forbidden band widths of $\text{TiO}_2(0.3\%)$ and $(0.05\text{M})\text{Ag}@\text{TiO}_2(0.3\%)$ of 2.92 eV and 2.73 eV. The forbidden band width calculated by the two methods is the same. Therefore, the absorption light range of $\text{TiO}_2(0.3\%)$ loaded with AgNPs moves toward the visible light direction, and the forbidden band width is reduced, which improves the light utilization efficiency of the sample.

We modified the content of Fig. 7 in the manuscript. Expressed in red font.

Introduction

(1) The starting phrase ‘ TiO_2 has very....international research’ may be either removed or condensed as it is too general in its content.

‘ TiO_2 has very....international research’ is indeed too general, and we decided to concentrate and modify this part. The modified content is indicated in red font in the

body of the manuscript. The changes are as follows:

Titanium dioxide (TiO_2) a multifunctional semiconductor metal oxide, which has attracted an extensive range of research attention because of their unique optical, electronic, and antibacterial properties. In practical applications, TiO_2 is not toxic, has super-hydrophilicity, and can completely contact with food[1] and does not affect human health. TiO_2 shows good development prospects in many fields. At present, TiO_2 has been widely used as a multifunctional material in the fields of solar cells[2], sensors[3], ceramics[4, 5], especially photocatalysts[6-8] in many environmental pollution control programs. Photocatalytic technology based on nano- TiO_2 materials provides a cheap, non-toxic, energy-efficient, and highly efficient method for degrading organic pollutants in air and water[9-12]. As an indirect bandgap semiconductor material, TiO_2 has typical semiconductor energy band characteristics. The energy band consists of a valence band (VB) filled with electron orbits, an empty orbital conduction band (CB) without electrons, and a band gap (E_g) between the valence band and the conduction band. When TiO_2 is not excited, electrons in the valence band do not automatically transition to the conduction band. Only when the energy excited by the photons is greater than E_g , the electrons in the valence band absorb the energy of the photon transition into the conduction band, and holes are generated in the valence band, "electron-hole pairs" (e^-h^+) are formed[13-16]. As the research groups continue to explore, the morphological structure of TiO_2 is also constantly changing. Nanorods, nanotubes, nanoflowers, nanoparticles and nanofibers have been studied and prepared. Although notable advances have been made, but the high recombination rate of the

photogenerated electron/hole pairs and the low utilization rate of ultraviolet hinders its further application in industry.

Section 3

(1) The ‘materials characterization’ must appear before ‘photocatalytic activity measurement’ to synchronize with ‘Results and discussion’ part!

Sorry, we misplaced the two parts of "material characterization" and "photocatalytic activity measurement". Thank you for your suggestion, we have completed the exchange of these two parts in the manuscript. For your convenience, the title ‘Materials characterization’ and ‘Photocatalytic activity measurement’ are indicated in red.

Section 4

(1) The ripening process leading to tubular structure and further cracking associated with oxalic acid should be properly addressed for more understanding.

Fig. 5 TG-DTA analysis of (a) H₂C₂O₄ and (b) TiO₂(0.3%) precursor fiber soaked in TBOT.

First, H₂C₂O₄ is added to the electrospinning solution. PS nanofibers with H₂C₂O₄ were prepared by electrospinning, and PS nanofibers were used as a sacrificial template for the preparation of TiO₂ nanotubes. During the heating and calcination process, the PS nanofibers and H₂C₂O₄ will completely decompose. In particular, H₂C₂O₄ releases a large amount of gas during the decomposition process. Fig. 5 shows that the complete decomposition temperature of H₂C₂O₄ (250 °C) is lower than the decomposition

temperature of PS, so under the same calcination conditions, $\text{H}_2\text{C}_2\text{O}_4$ will preferentially decompose. The presence of a large amount of gas causes the PS fiber to loosen and accelerate the decomposition of PS. At the same time, 350~450 °C is the key period for the fixed shape of TiO_2 nanotubes. The impact of a large amount of gas (H_2O and CO_2 produced by the decomposition of PS) during the critical period of TiO_2 nanotubes will cause the TiO_2 nanotubes to crack, forming a pod-shaped TiO_2 NBTs. Moreover, the surface of TiO_2 NBTs will have a large number of gas overflow holes.

(2) Is it possible to provide TGA/DTA analysis which might provide more information on the decomposition of additives?

Thanks to the reviewer for this suggestion. In fact, we have already done this part of the work, but it is not in the body of the manuscript, which affects your understanding and reading of the content of the article. Therefore, we decided to put this part of the work in the body. The following is a TG/DTA analysis of pure $\text{H}_2\text{C}_2\text{O}_4$ and sample $\text{TiO}_2(0.3\%)$:

Fig. 6 TG-DTA analysis of (a) $\text{H}_2\text{C}_2\text{O}_4$ and (b) $\text{TiO}_2(0.3\%)$ precursor fiber soaked in TBOT.

TG-DTA analysis of pure $\text{H}_2\text{C}_2\text{O}_4$ and PS/TBOT composite fiber samples with $\text{H}_2\text{C}_2\text{O}_4$ content of 0.3%, the result was shown in Fig. 6. As can be seen from Fig. 6(a), the complete decomposition temperature of pure $\text{H}_2\text{C}_2\text{O}_4$ is 250 °C, and $\text{H}_2\text{C}_2\text{O}_4$ has been

completely decomposed when PS has not been decomposed. Fig. 6(b) shows that the thermal decomposition process of the sample was divided into three stages. The first stage occurred between 70°C~350°C. The sample lost water and H₂C₂O₄ decomposed to release CO₂ gas. The second stage is 350~450°C, which represents the decomposition of PS organic components and some of the organics produced by TBOT hydrolysis. The third stage is 450°C~700°C, mainly including PS main chain degradation and amorphous TiO₂ to anatase phase two processes.

(3) Comment on the amount of Ag metal leaching (if any) from the catalyst surface at the end of the reaction.

Fig. 7 (0.05M)Ag@TiO₂(0.3%) and RhB solution mixture in quartz tube at the end of reaction (a), recover (0.05M)Ag@TiO₂(0.3%) sample picture (b).

Sorry, we did not discuss the amount of Ag metal leaching on the catalyst surface at the end of the reaction. Because it is difficult to separate the Ag metal from the Ag@TiO₂ catalyst. In the catalytic degradation experiment, in order to allow the Ag@TiO₂ catalyst to be in full contact with the RhB solution, we chose to stir the Ag@TiO₂ catalyst and the RhB mixture at high rotation speed. However, during the high-speed agitation, a small amount of AgNPs will fall off the surface of the TiO₂ NBTs. Fig. 7(a) is a photograph after the end of the reaction, the Ag metal and the Ag@TiO₂ catalyst

are thoroughly mixed. It can also be observed from Fig. 7(b) that the recovered Ag metal and Ag@TiO₂ catalyst have been completely mixed and cannot be separated. Therefore, it is difficult to determine the amount of Ag metal leaching on the surface of the catalyst at the end of the reaction.

Thanks again to the reviewers for their constructive comments and suggestions. We have reworked the content of the electronic supplementary material to make up for the deficiencies in the manuscript.

Finally, we sincerely appreciate your insightful and constructive comments and suggestions. We appreciate for Editor/Reviewers' warm work earnestly, and hope that the correction will meet with approval.

Once again, thank you very much for taking the time to review this paper.

Sincerely,

Shang Wang